# Transport with Support: Data-Conditional Diffusion Bridges

## Abstract

The dynamic Schrödinger bridge problem provides an appealing setting for posing optimal transport problems as learning non-linear diffusion processes and enables efficient iterative solvers. Recent works have demonstrated state-of-the-art results (*e.g.*, in modelling single-cell embryo RNA sequences or sampling from complex posteriors) but are typically limited to learning bridges with only initial and terminal constraints. Our work extends this paradigm by proposing the Iterative Smoothing Bridge (ISB). We combine learning diffusion models with Bayesian filtering and optimal control, allowing for constrained stochastic processes governed by sparse observations at intermediate stages and terminal constraints. We assess the effectiveness of our method on synthetic and real-world data and show that the ISB generalises well to high-dimensional data, is computationally efficient, and provides accurate estimates of the marginals at intermediate and terminal times.

## 1 Introduction

Generative diffusion models have gained increasing popularity and achieved impressive results in a variety of challenging application domains, such as computer vision (*e.g.*, Ho et al., 2020; Song et al., 2021a; Dhariwal & Nichol, 2021), reinforcement learning (*e.g.*, Janner et al., 2022), and time series modelling (*e.g.*, Rasul et al., 2021; Vargas et al., 2021; Tashiro et al., 2021; Park et al., 2022). Recent works have explored connections between denoising diffusion models and the dynamic Schrödinger bridge problem (SBP, *e.g.*, Vargas et al., 2021; De Bortoli et al., 2021; Shi et al., 2022) to adopt iterative schemes for solving the dynamic optimal transport problem more efficiently. The solution of the SBP that correspond to denoising diffusion models is then given by the finite-time process, which is the closest in Kullback–Leibler (KL) divergence to the forward noising process of the diffusion model under marginal constraints. Data is then generated by time-reversing the process.

In many applications, the interest is not purely in modelling transport between an initial and terminal state distribution In naturally occurring generative processes, we typically observe snapshots of realizations *along intermediate stages* of individual sample trajectories (see Fig. 1). Such problems arise in medical diagnosis (*e.g.*, tissue changes and cell growth), demographic modelling, environmental dynamics, and animal movement modelling—see Fig. 4 for modelling bird migration and wintering patterns. Recently, constrained optimal control problems have been explored by adding additional fixed path constraints (Maoutsa et al., 2020; Maoutsa & Opper, 2021) or modifying the prior processes (Fernandes et al., 2021). However, defining meaningful fixed path constraints or prior processes for the optimal control problems can be challenging, while sparse observational data are accessible in many real-world applications.

In this work, we propose the *Iterative Smoothing Bridge* (ISB), an iterative method for solving control problems under sparse observational data constraints and constraints on the initial and terminal distribution. We perform the conditioning by leveraging the iterative pass idea from the Iterative Proportional Fitting procedure (IPFP) (Kullback, 1968; De Bortoli et al., 2021) procedure and applying differentiable particle filtering (Reich, 2013; Corenflos et al., 2021) within the outer loop. Integrating sequential Monte Carlo methods (*e.g.*, Doucet et al., 2001; Chopin & Papaspiliopoulos, 2020) into the IPFP framework in such a way is non-trivial and can be understood as a novel iterative version of the algorithm by Maoutsa & Opper (2021) but with more general terminal constraints and path constraints defined by data.

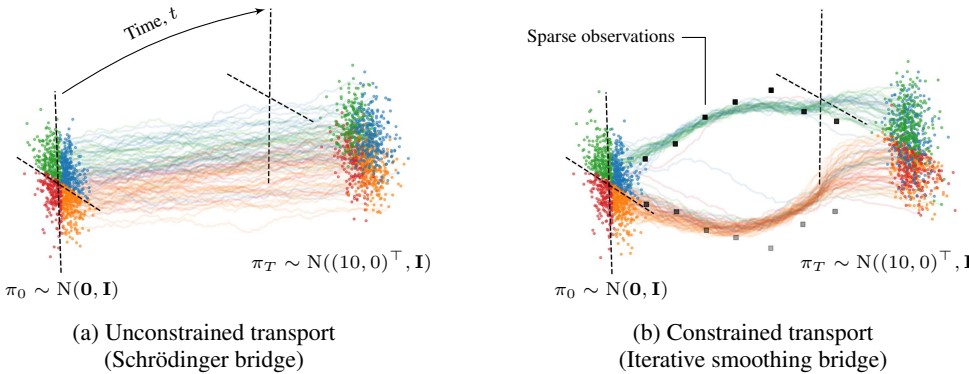

Figure 1: Illustrative example transport between an initial unit Gaussian and a shifted unit Gaussian at the terminal time $T$. Unconstrained transport on the left and the solution constrained by sparse observations (■) on the right. Colour coding of the initial points is only for distinguishing the paths.

We summarize the contributions as follows. *(i)* We propose a novel method for solving constrained optimal transport as a bridge problem under sparse observational data constraints. *(ii)* Thereof, we utilize the strong connections between the constrained bridging problem and particle filtering in sequential Monte Carlo, extending them from pure inference to learning. Additionally, *(iii)* we demonstrate practical efficiency and show that the iterative smoothing bridge approach scales to high-dimensional data.

## 1.1 RELATED WORK

**Schrödinger bridges** The problem of learning a stochastic process moving samples from one distribution to another can be posed as a type of a transport problem known as a dynamic Schrödinger bridge problem (SBP, *e.g.*, Schrödinger, 1932; Léonard, 2014), where the resulting marginal densities are desired to resemble a given reference measure. In machine learning literature, the problem has been studied through learning the drift function of the dynamical system (De Bortoli et al., 2021; Wang et al., 2021; Vargas et al., 2021; Bunne et al., 2022). When an SDE system also defines the reference measure, the bridge problem becomes synonymous with a constrained optimal control problem (*e.g.*, Caluya & Halder, 2022; 2021; Chen et al., 2021), which has been leveraged in learning Schrödinger bridges by Tianrong Chen (2022) through forward–backward SDEs. An optimal control problem with both constraints on the initial and terminal distribution and a fixed path constraint has been studied in Maoutsa et al. (2020) and Maoutsa & Opper (2021), where particle filtering is applied to continuous path constraints but the boundary constraints are defined by a single point. Furthermore, the combination of Schrödinger bridges and state-space models has been studied by Reich (2019), in a setting where Schrödinger bridges are applied to the transport problem between filtering distributions.

**Diffusion models in machine learning** The recent advances in diffusion models in machine learning literature have been focused in generating samples from complex distributions defined by data through transforming samples from an easy-to-sample distribution by a dynamical system (*e.g.*, Ho et al., 2020; Song et al., 2021b;a; Nichol & Dhariwal, 2021). The concept of reversing SDE trajectories via score-based learning (Hyvärinen & Dayan, 2005; Vincent, 2011) has allowed for models scalable enough to be applied to high-dimensional data sets directly in the data space. In earlier work, score-based diffusion models have been applied to problems where the dynamical system itself is of interest, for example, for the problem of time series amputation in Tashiro et al. (2021) and inverse problems in imaging in Song et al. (2022). Other dynamical models parametrized by neural networks have been applied to modelling latent time-series based on observed snapshots of dynamics (Rubanova et al., 2019; Li et al., 2020), but without further constraints on the initial or terminal distributions.

**State-space models** In their general form, state-space models combine a latent space dynamical system with an observation (likelihood) model. Evaluating the latent state distribution based on observational data can be performed by applying particle filtering and smoothing (Doucet et al., 2000) or by approximations of the underlying state distribution of a non-linear state-space model by a specific model family, for instance, a Gaussian (see Särkkä, 2013, for an overview). Speeding up parameter inference and learning in state-space models has been widely studied (*e.g.*, Schön

et al., 2011; Svensson & Schön, 2017; Kokkala et al., 2014). Particle smoothing can be connected to Schrödinger bridges via the two-filter smoother (*e.g.*, Bresler, 1986; Briers et al., 2009; Hostettler, 2015), where the smoothing distribution is estimated by performing filtering both forward from the initial constraint and backward from the terminal constraint. We refer to Mitter (1996) and Todorov (2008) for a more detailed discussion on the connection of stochastic control and filtering and to Chopin & Papaspiliopoulos (2020) for an introduction to particle filters.

## 2 BACKGROUND

Let $\mathcal{C} = C([0, T], \mathbb{R}^d)$ denote the space of continuous functions from $[0, T]$ to $\mathbb{R}^d$ and let $\mathcal{B}(\mathcal{C})$ denote the Borel $\sigma$-algebra on $\mathcal{C}$. Let $\mathscr{P}(\pi_0, \pi_T)$ denote the space of probability measures on $(\mathcal{C}, \mathcal{B}(\mathcal{C}))$ such that the marginals at $0, T$ coincide with probability densities $\pi_0$ and $\pi_T$, respectively. The KL divergence from measure $\mathbb{Q}$ to measure $\mathbb{P}$ is written as $\mathrm{D_{KL}}\left[\mathbb{Q} \,\|\, \mathbb{P}\right]$, where we assume that $\mathbb{Q} \ll \mathbb{P}$. For modelling the time dynamics, we assume a (continuous-time) state-space model consisting of a non-linear latent Itô SDE (see, *e.g.*, Øksendal, 2003; Särkkä & Solin, 2019) in $[0, T] \times \mathbb{R}^d$ with drift function $f_\theta(\cdot)$ and diffusion function $g(\cdot)$, and a Gaussian observation model, *i.e.*,

$$\mathbf{x}_0 \sim \pi_0, \quad \mathrm{d}\mathbf{x}_t = f_\theta(\mathbf{x}_t, t)\,\mathrm{d}t + g(t)\,\mathrm{d}\boldsymbol{\beta}_t, \quad \mathbf{y}_k \sim \mathrm{N}(\mathbf{y}_k \,|\, \mathbf{x}_t, \sigma^2\,\mathbf{I}_d)\big|_{t=t_k}, \tag{1}$$

where the drift $f_\theta : \mathbb{R}^d \times [0, T] \to \mathbb{R}^d$ is a mapping modelled by a neural network (NN) parameterized by $\theta \in \Theta$, diffusion $g : [0, T] \to \mathbb{R}$ and $\boldsymbol{\beta}_t$ denotes standard $d$-dimensional Brownian motion. $\mathbf{x}_t$ denotes the latent stochastic process and $\mathbf{y}_t$ denotes the observation-space process. In practice, we consider the continuous-discrete time setting, where the process is observed at discrete time instances $t_k$ such that observational data can be given in terms of a collection of input–output pairs $\{(t_k, \mathbf{y}_k)\}_{k=1}^M$.

### 2.1 SCHRÖDINGER BRIDGES AND OPTIMAL CONTROL

The Schrödinger bridge problem (SBP, Schrödinger, 1932; Léonard, 2014) can be described as an entropy-regularized optimal transport problem where the optimality is measured through the KL divergence from a reference measure $\mathbb{P}$ to the posterior measure $\mathbb{Q}$, with fixed initial and final densities $\pi_0$ and $\pi_T$, *i.e.*,

$$\min_{\mathbb{Q} \in \mathscr{P}(\pi_0, \pi_T)} \mathrm{D_{KL}}\left[\mathbb{Q} \,\|\, \mathbb{P}\right]. \tag{2}$$

In this work, we consider only the case where the measures $\mathbb{P}$ and $\mathbb{Q}$ are constructed as the marginals of an SDE, *i.e.*, $\mathbb{Q}_t$ is the probability measure of the marginal of the SDE in Eq. (1) at time $t$, whereas $\mathbb{P}_t$ corresponds to the probability measure of the marginal of a reference SDE $\mathrm{d}\mathbf{x}_t = f(\mathbf{x}_t, t)\,\mathrm{d}t + g(t)\,\mathrm{d}\boldsymbol{\beta}_t$, at time $t$, where we call $f$ the reference drift. Under the optimal control formulation of the SBP (Caluya & Halder, 2021) the KL divergence in Eq. (2) reduces to

$$\mathbb{E}\left[\int_0^T \frac{1}{2g(t)^2}\|f_\theta(\mathbf{x}_t, t) - f(\mathbf{x}_t, t)\|^2 \,\mathrm{d}t\right], \tag{3}$$

where the expectation is over paths from Eq. (1). Rüschendorf & Thomsen (1993) and Ruschendorf (1995) showed that a solution to the SBP can be obtained by iteratively solving two half-bridge problems using the Iterative Proportional Fitting procedure (IPFP) for $l = 0, 1, \dots, L$ steps:

$$\mathbb{Q}_{2l+1} = \underset{\mathbb{Q} \in \mathscr{P}(\cdot, \pi_T)}{\arg\min}\, \mathrm{D_{KL}}\left[\mathbb{Q} \,\|\, \mathbb{Q}_{2l}\right] \quad \text{and} \quad \mathbb{Q}_{2l+2} = \underset{\mathbb{Q} \in \mathscr{P}(\pi_0, \cdot)}{\arg\min}\, \mathrm{D_{KL}}\left[\mathbb{Q} \,\|\, \mathbb{Q}_{2l+1}\right], \tag{4}$$

where $\mathbb{Q}_0$ is set as the reference measure, and $\mathscr{P}(\pi_0, \cdot)$ and $\mathscr{P}(\cdot, \pi_T)$ denote the sets of probability measures with only either the marginal at time 0 or time $T$ coinciding with $\pi_0$ or $\pi_T$, respectively. Recently the IPFP to solving Schrödinger bridges has been adapted as a machine learning problem (Bernton et al., 2019; Vargas et al., 2021; De Bortoli et al., 2021). In practice, the interval $[0, T]$ is discretized and the forward drift $f_\theta$ and the backward drift $b_\phi$ of the corresponding reverse-time process (Haussmann & Pardoux, 1986; Föllmer, 1988) are modelled by NNs. Under the Gaussian transition approximations, the resulting discrete-time diffusion model can be reversed by applying a mean-matching based objective.

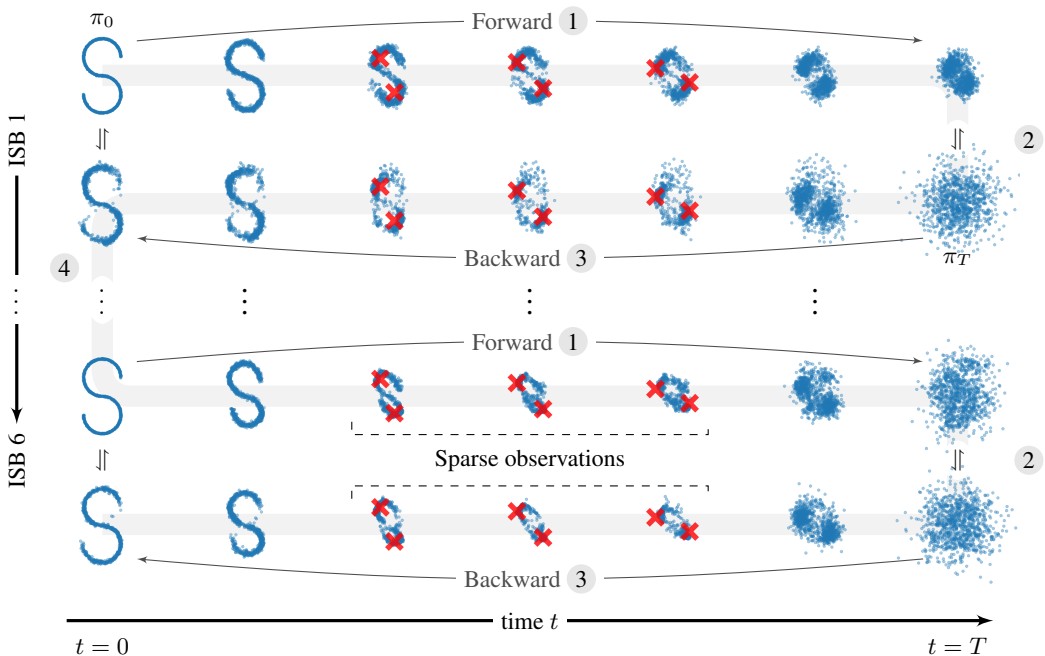

Figure 2: Sketch of a diffusion bridge between a 2D data distribution ($\pi_0$) and an isotropic Gaussian ($\pi_T$) constrained by sparse observations (✖). The forward diffusion at the first iteration (ISB 1) learns to account for the sparse observations but does not converge to the correct terminal distribution ($t = T$), and the backward diffusion *vice versa*. After iterating (ISB 6), the forward and backward diffusions converge to the correct targets and are able to account for the sparse observational data.

## 3  METHODS

Given an initial and terminal distribution $\pi_0$ and $\pi_T$, we are interested in learning a data-conditional bridge between $\pi_0$ and $\pi_T$. Let $\mathcal{D} = \{(t_j, \mathbf{y}_j)\}_{j=1}^M$ be a set of $M$ sparsely observed values, *i.e.*, only a few or no observations are made at each point in time, and let the state-space model of interest be given by Eq. (1). Note that we deliberately use $(t_j, \mathbf{y}_j)$ (instead of $(t_k, \mathbf{y}_k)$) to highlight that we allow for multiple observations at the same time point $t_k$. Our aim is to find a parameterization of the drift function $f_\theta$ such that evolving $N$ particles $\mathbf{x}_t^i$, with $\mathbf{x}_0^i \sim \pi_0$ (with $i = 1, 2, \dots, N$), according to Eq. (1) will result in samples $\mathbf{x}_T^i$ from the terminal distribution $\pi_T$. Inspired by the IPFP by De Bortoli et al. (2021), which decomposes the SBP into finding two half-bridges, we propose to iteratively solve the two half-bridge problems while accounting for the additional sparse observations simultaneously. For this, let

$$d\mathbf{x}_t = f_{l,\theta}(\mathbf{x}_t, t)\,dt + g(t)\,d\boldsymbol{\beta}_t, \qquad\qquad \mathbf{x}_0 \sim \pi_0, \qquad (5)$$

$$d\mathbf{z}_t = b_{l,\phi}(\mathbf{z}_t, t)\,dt + g(t)\,d\hat{\boldsymbol{\beta}}_t, \qquad\qquad \mathbf{z}_0 \sim \pi_T, \qquad (6)$$

denote the forward and backward SDE at iteration $l = 1, 2, \dots, L$, where $\hat{\boldsymbol{\beta}}_t$ is the reverse-time Brownian motion. For simplicity, we denote $\boldsymbol{\beta}_t = \hat{\boldsymbol{\beta}}_t$ when the direction of the SDE is clear.

To learn the data-conditioned bridge, we iteratively employ the following steps: ① evolve and filter *forward* particle trajectories according to Eq. (5) with drift $f_{l-1,\theta}$ and observations $\{(t_k, \mathbf{y}_k)\}_{k=1}^M$, ② learn the drift function $b_{l,\phi}$ for the reverse-time SDE, ③ evolve and filter *backward* particle trajectories according to Eq. (6) with the drift $b_{l,\phi}$ learned in step ② and observations $\{(t_k, \mathbf{y}_k)\}_{k=1}^M$, and ④ learn the drift function $f_{l,\theta}$ for the forward SDE based on the backward particles. Fig. 2 illustrates the forward and backward process of our iterative scheme for a data-conditioned denoising diffusion bridge. Next, we will go through steps ①–④ in detail and introduce the Iterative Smoothing Bridge method for solving data-conditional diffusion bridges.

### 3.1 THE ITERATIVE SMOOTHING BRIDGE

The Iterative Smoothing Bridge (ISB) method iteratively generates particle filtering trajectories (steps ①and ③ in Fig. 2) and learns the parameterizations of the forward and backward drift functions $f_{l,\theta}$ and $b_{l,\phi}$ (steps ② and ④) by minimizing a modified version of the mean-matching objective presented by De Bortoli et al. (2021). Note that steps ② and ④ are dependent on applying differential resampling in the particle filtering steps ① and ③ for reversing the generated trajectories. We will now describe the forward trajectory generating step ① and the backward drift learning step ② in detail. Steps ③ and ④ are given by application of ① and ② on their reverse-time counterparts.

**Step ① (and ③):** Given a fixed discretization of the time interval $[0, T]$ denoted as $\{t_k\}_{k=1}^K$ with $t_1 = 0$ and $t_K = T$, denote the time step lengths as $\Delta_k = t_{k+1} - t_k$. By truncating the Itô–Taylor series of the SDE, we can consider an Euler–Maruyama (*e.g.*, Ch. 8 in Särkkä & Solin, 2019) type of discretization for the continuous-time problem. We give the time-update of the $i^{\text{th}}$ particle at time $t_k$ evolved according to Eq. (5) before conditioning on the observational data as

$$\tilde{\mathbf{x}}_{t_k}^i = \mathbf{x}_{t_{k-1}} + f_{l-1,\theta}(\mathbf{x}_{t_{k-1}, t_{k-1}})\Delta_k + g(t_{k-1})\sqrt{\Delta_k}\,\boldsymbol{\xi}_k^i, \tag{7}$$

where $\boldsymbol{\xi}_k^i \sim \mathrm{N}(\mathbf{0}, \mathbf{I})$. In step ③, the particles $\tilde{\mathbf{z}}_{t_k}^i$ of the backward SDE Eq. (6) are similarly obtained. The SDE dynamics sampled in steps ① and ③ apply the learned drift functions $f_{l-1,\theta}$ and $b_{l,\phi}$ from the previous step and do not require sampling from the underlying SDE model. For times $t_k$ at which no observations are available, we set $\mathbf{x}_t^i = \tilde{\mathbf{x}}_t^i$ (and $\mathbf{z}_{t_k}^i = \tilde{\mathbf{z}}_{t_k}^i$ respectively) and otherwise compute the particle filtering weights $w_{t_k}^i$ based on the observations $\{(t_j, \mathbf{y}_j) \in \mathcal{D} \,|\, t_j = t_k\}$ for resampling. See Sec. 3.2 for details on the particle filtering proposal density and calculation of the particle weights.

For resampling, we employ a *differentiable resampling* procedure, where the particles and weights $(\tilde{\mathbf{x}}_{t_k}^i, w_{t_k}^i)$ are transported to uniformly weighted particles $\mathbf{x}_{t_k}^i$ by solving an entropy-regularized optimal transport problem (Cuturi, 2013; Peyré & Cuturi, 2019; Corenflos et al., 2021), see App. D for further details. Through application of the $\varepsilon$-regularized optimal transport map $\mathbf{T}_{(\varepsilon)} \in \mathbb{R}^{N \times N}$ (see Corenflos et al., 2021, for details) the particles are resampled via the map to $\mathbf{x}_{t_k}^i = \tilde{\mathbf{X}}_{t_k}^\top \mathbf{T}_{(\varepsilon),i}$, where $\tilde{\mathbf{X}}_{t_k} \in \mathbb{R}^{N \times d}$ denotes the stacked particles $\{\tilde{\mathbf{x}}_{t_k}^i\}_{i=1}^N$ at time $t_k$ before resampling. The resampled particles for the backward process are given similarly.

**Step ② (and ④):** Given the particles $\{\mathbf{x}_{t_k}^i\}_{k=1, i=1}^{K, N}$, we now aim to learn the drift function for the respective reverse-time process. In case no observation is available at time $t_k$, we apply the mean-matching loss based on a Gaussian transition approximation proposed in De Bortoli et al. (2021):

$$\ell_{k+1,\text{no obs}}^i = \|b_{l,\phi}(\mathbf{x}_{t_{k+1}}^i, t_{k+1})\Delta_k - \mathbf{x}_{t_{k+1}}^i - f_{l-1,\theta}(\mathbf{x}_{t_{k+1}}^i, t_k)\Delta_k + \mathbf{x}_{t_k}^i + f_{l-1,\theta}(\mathbf{x}_{t_k}^i, t_k)\Delta_k\|^2. \tag{8}$$

In case an observation is available at time $t_k$ the particle values $\tilde{\mathbf{X}}_{t_k}$ will be coupled through the optimal transport map. Therefore, the transition density is a sum of Gaussian variables (see App. A for details and a derivation), and the mean-matching loss is therefore given by

$$s\ell_{k+1,\text{obs}}^i = \|b_{l,\phi}(\mathbf{x}_{t_{k+1}}^i, t_{k+1})\Delta_k - \mathbf{x}_{t_{k+1}}^i - f_{l-1,\theta}(\mathbf{x}_{t_{k+1}}^i, t_k)\Delta_k$$
$$+ \textstyle\sum_{n=1}^N T_{(\varepsilon),i,n}\left(\mathbf{x}_{t_k}^n + f_{l-1,\theta}(\mathbf{x}_{t_k}^n, t_k)\Delta_k\right)\|^2. \tag{9}$$

The overall objective function is a combination of both loss functions, with the respective mean-matching loss depending on whether $t_k$ is an observation time. The final loss function is written as:

$$\ell(\phi) = \textstyle\sum_{i=1}^N \left[\sum_{k=1}^K \ell_{k,\text{obs}}^i(\phi)\mathbb{I}_{y_{t_k} \neq \emptyset} + \sum_{k=1}^K \ell_{k,\text{no obs}}^i(\phi)\mathbb{I}_{y_{t_k} = \emptyset}\right], \tag{10}$$

where $\mathbb{I}_{\text{cond.}}$ denotes an indicator function that returns '1' iff the condition is true, and '0' otherwise. Consequently, the parameters $\phi$ of $b_{l,\phi}$ are learned by minimizing Eq. (10) through gradient descent. In practice, a cache of trajectories $\{\mathbf{x}_{t_k}^i\}_{k=1, i=1}^{K, N}$ is maintained through training of the drift functions, and refreshed at fixed number of inner loop iterations, as in De Bortoli et al. (2021), avoiding differentiation over the SDE generation computational graph. The calculations for step ④ follow similarly.

The learned backward drift $b_{l,\phi}$ can be interpreted as an analogy of the backward drift in Maoutsa & Opper (2021), connecting our approach to solving optimal control problems through Hamilton–Jacobi equations, see App. A.2 for an analysis of the backwards SDE and the control objective. While we are generally considering problem settings where the number of observations is low, we propose that letting $M \to \infty$ yields the underlying marginal distribution, see Prop. 2 in App. A.3.

### 3.2 COMPUTATIONAL CONSIDERATIONS

The ISB algorithm is a generic approach to learn data-conditional diffusion bridges under various choices of, *e.g.*, the particle filter proposal density or the reference drift. Next, we cover practical considerations for the implementation of the method and highlight the model choices in the experiments.

**Multiple observations per time step**  Naturally, we can make more than one observation at a single point in time $t_k$, denoted as $\mathcal{D}_{t_k} = \{(t_j, \mathbf{y}_j) \in \mathcal{D} \mid t_j = t_k\}$. To compute particle weights $w_{t_k}^i$ for the $i^{\text{th}}$ particle we consider only the $H$-nearest neighbours of $\mathbf{x}_{t_k}^i$ in $\mathcal{D}_{t_k}$ instead of all observations in $\mathcal{D}_{t_k}$. By restricting to the $H$-nearest neighbours, denoted as $\mathcal{D}_{t_k}^H$, we introduce an additional locality to the proposal density computation which can be helpful in case of multimodality. On the other hand, letting $H > 1$ results in weights which take into account the local density of the observations, not only the distance to the nearest neighbour. In experiments with few observations, we set $H = 1$, the choice of $H$ is discussed when we have set the value higher.

**Particle filtering proposal**  The proposal density chosen for the ISB is the bootstrap filter, where the proposal matches the Gaussian transition density $p(\mathbf{x}_{t_k} \mid \mathbf{x}_{t_{k-1}})$. Assuming a Gaussian noise model $\mathrm{N}(\mathbf{0}, \sigma^2 \mathbf{I})$, the unnormalized log-weights for the $i^{\text{th}}$ particle at time $t_k$ are given by:

$$\log w_{t_k}^i = -\frac{1}{2\sigma^2} \sum_{\mathbf{y}_j \in \mathcal{D}_{t_k}^H} \|\mathbf{x}_{t_k}^i - \mathbf{y}_j\|^2. \tag{11}$$

**Observational noise schedule**  In practice, using a constant observation noise $\sigma^2$ variance can result in an iterative scheme which does not have a stationary point as $L \to \infty$. Even if the learned drift function $f_{l,\theta}$ was optimal, the filtering steps ① and ③ would alter the trajectories unless all particles would have uniform weights. Thus, we introduce a noise schedule $\kappa(l)$ which ensures that the observation noise increases in the number of ISB iterations, causing ISB to converge to the IPFP (De Bortoli et al., 2021) as $L \to \infty$. We found that letting the observation noise first decrease and then increase (in the spirit of simulated annealing) often outperformed a strictly increasing observation noise schedule. The noise schedule is studied in App. C, where we derive the property that letting $L \to \infty$ yields IPFP.

**Drift initialization**  Depending on the application, one may choose to incorporate additional information by selecting an appropriate initial drift. A possible choice includes a pre-trained neural network drift learned to transport $\pi_0$ to $\pi_T$ without accounting for observations. However, starting from a drift for the unconstrained SBP can be problematic in cases where the observations are far away from the unconstrained bridge. To encouraged exploration, one may choose $f_0 = 0$ for the initial drift. In various problem settings, we found both, a zero drift and starting from the SBP, to be successful in the experiments, see App. C for discussion.

## 4 EXPERIMENTS

To assess the properties and performance of the ISB, we present a range of experiments that demonstrate how the iterative learning procedure can incorporate both observational data and terminal constraints. We start with simple examples that build intuition (*cf.* Fig. 1 and Fig. 2) and show standard ML benchmark tasks. For quantitative assessment, we design an experiment with a non-linear SDE for which the marginal distributions are available in closed-form. Finally, we demonstrate our model both in a highly multimodal bird migration task, conditioned image generation, and in a single-cell embryo RNA modelling problem. Ablation studies are found in App. C.

**Experiment setup**  In all experiments, the forward and backward drift functions $f_\theta$ and $b_\phi$ are parametrized as neural networks. For low-dimensional experiments we apply the MLP block design as in De Bortoli et al. (2021), and for the image experiment an U-Net as in Nichol & Dhariwal (2021). The latent state SDE was simulated by Euler–Maruyama with a fixed time-step of $0.01$ over $100$ steps and $1000$ particles if not otherwise stated. All low-dimensional (at most $d = 5$) experiments were run on a MacBook Pro laptop CPU, whereas the image experiments used a single Nvidia A100 GPU and ran for 5 h 10 min. Notice that since ISB only performs particle filtering outside the stochastic gradient training loop, the training runtime is in the same order as in the earlier Schrödinger bridge image generation experiments of De Bortoli et al. (2021). Thus we omit any wall-clock timings. Full details for all the experiments are included in App. B.

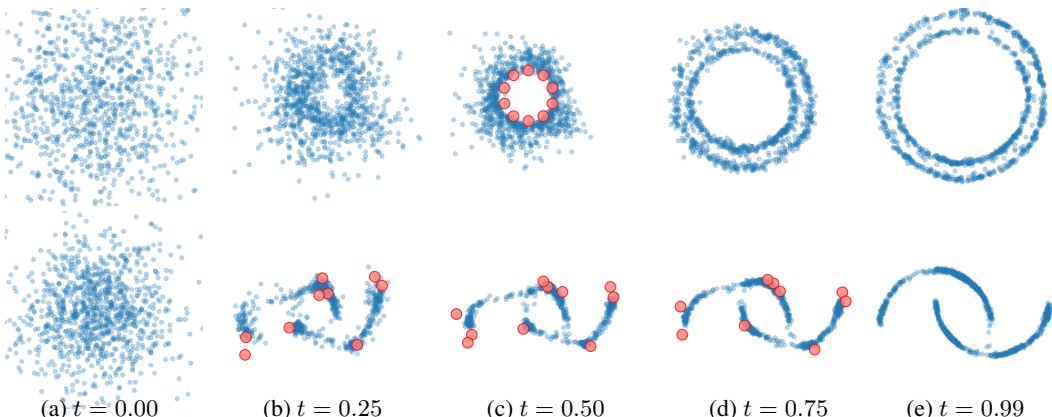

(a) $t = 0.00$     (b) $t = 0.25$     (c) $t = 0.50$     (d) $t = 0.75$     (e) $t = 0.99$

Figure 3: 2D toy experiments from scikit-learn with both cases starting from a Gaussian: The TWO CIRCLES (top) and TWO MOONS (bottom) data sets, with observations (red markers) constraining the problem. For the circles, the 10 circular observations at $t = 0.5$ first force the method to create a circle that then splits into two; in the lower plot the observations at $t \in [0.25, 0.5, 0.75]$ split the data into clusters before joining them into two moons. See Fig. 6 in the Appendix for the IPFP result.

All experiment settings include a number of hyperparameter choices, some of them typical to all diffusion problems and some specific to particle filtering and smoothing. The diffusion $g(t)$ is a pre-determined function not optimized in the training. We divide the experiments to two main subsets: problems of 'sharpening to achieve a data distribution' and 'optimal transport problems'. In the former, the initial distribution has a support overlapping with the terminal distribution and the process noise level $g(t)$ goes from high to low as time progresses. Conversely in the latter setting, the particles sampled from the initial distribution must travel to reach the support of the terminal distribution, and we chose to use a constant process noise level. Perhaps the most significant choice of hyperparameter is the observational noise level, as it imposes a preference on how closely should the observational points be followed, see App. C.1 for details.

**2D toy examples** We show illustrative results for the TWO MOONS and CIRCLES from scikit-learn. We add artificial observation data to bias the processes. For the circles, the observational data consists of 10 points, spaced evenly on the circle. The points are all observed simultaneously, at halfway through the process, forcing the marginal density of the generating SDE to collapse to the small circle, and then to expand. For the two moons, the observational data is collected from 10 trajectories of a diffusion model which generates the two moons from noise, and these 10 trajectories are then observed at three points in time. Results are visualized in Fig. 3 (see videos in supplement). For reference, we have included plots of the IPFP dynamics in the supplement, see Fig. 6.

**Quantitative comparison on the Beneš SDE**
In order to quantify how observing a process in between its initial and terminal states steers the ISB model to areas with higher likelihood, we test its performance on a Beneš SDE model (see, *e.g.* Särkkä & Solin, 2019). The Beneš SDE is a non-linear one-dimensional SDE of form

Table 1: Results for the Beneš experiment.

| METHOD | Negative log predictive density | | |
| --- | --- | --- | --- |
| | AVERAGE | MIDDLE | END |
| Schrödinger B | 4.787 | 3.565 | 0.1919 |
| Iterative smoothing B | **3.557** | **2.985** | **0.1567** |

$\mathrm{d}x_t = \tanh(x_t)\,\mathrm{d}t + \mathrm{d}\beta_t$ with $x_0 = 0$, but its marginal density is available in closed-form, allowing for negative log-likelihood evaluation. We simulate trajectories from the Beneš SDE and from the reverse drift and stack the reversed trajectories. The terminal distribution is shifted and scaled so that the Beneš SDE itself does not solve the transport problem from $\pi_0$ to $\pi_T$, see App. B.2 for details and visualizations of the processes.

As a baseline, we fit a Schrödinger bridge model with no observational data, using the Beneš SDE drift as the reference model. The ISB model is initialized with a zero-drift model (not with the Beneš as reference), thus making learning more challenging. We compare the models in terms of negative log predictive density in Table 1, where we see that the ISB model captures the process well on average (over the entire time-horizon) and at selected marginal times.

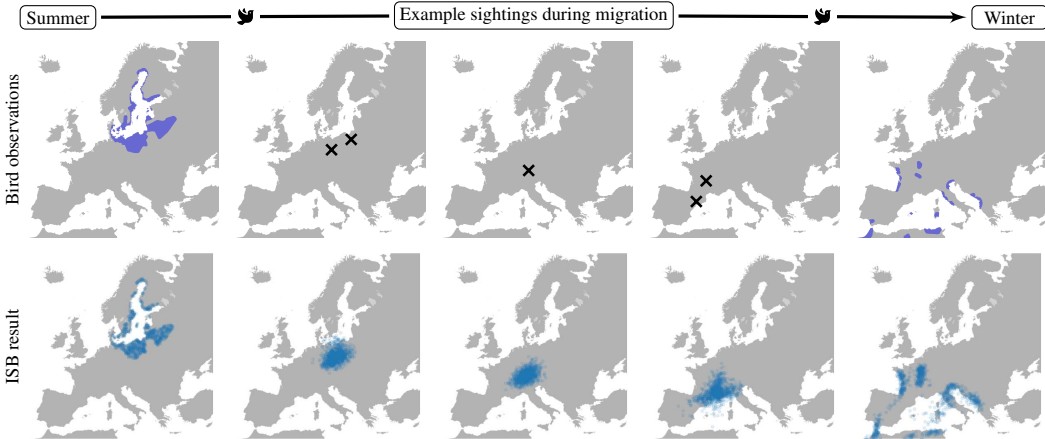

Figure 4: Bird migration example. The top row describes nesting and wintering areas for the birds as well as example sightings during migration. At the bottom, we show the marginal densities of the ISB model from the initial to terminal distribution that match the sightings along the migration.

**Bird migration** Bird migration can be seen as a regular seasonal transport problem, where birds move (typically North–South) along a flyway, between breeding and wintering grounds. We take this as a motivating example of constrained optimal transport, where the geographical and constraints and preferred routes are accounted for by bird sighting data (see Fig. 4 top). By adapting data from Ambrosini et al. (2014) and Pellegrino et al. (2015), we propose a simplified data set for geese migration in Europe (OIBMD: ornithologically implausible bird migration data; available in the supplement). We applied the ISB for 12 iterations, with a linear observation noise schedule from 1 to 0.2, and constant diffusion noise 0.05. The drift function was initialized as a zero-function, and thus the method did not rely on a separately fit model optimized for generating the wintering distribution based on the breeding distribution. For comparison, we include the Schrödinger bridge results in App. B.3.

**Constraining an image generation process** We demonstrate that the ISB approach scales well to high-dimensional inputs by studying a proof-of-concept image generation task. We modify the diffusion generative process of the MNIST (LeCun et al., 1998) digit 8 by artificial observations steering the dynamical system in the middle of the generation process. While the concept of observations in case of image generation is somewhat unnatural, it showcases the scalability of the method to high-dimensional data spaces. Here, the drift is initialized using a pre-trained neural network obtained by first running a Schrödinger bridge model for image generation. The process is then given an observation in the form of a bottom-half of a MNIST digit 8 in the middle of the dynamical process. As the learned model uses information from the observation both before and after the observation time, the lower half of the image is sharper than the upper half. We provide further details on this experiment and sampled trajectories in App. B.4.

**Single-cell embryo RNA-seq** Lastly, we evaluated our approach on an Embryoid body scRNA-seq time course (Tong et al., 2020). The data consists of RNA measurements collected over five time ranges from a developing human embryo system. No trajectory information is available, instead we only have

Table 2: Results for single-cell embryo RNA experiment.

| METHOD | Earth mover's distance | | | | |
| --- | --- | --- | --- | --- | --- |
| | $t{=}0$ | $t{=}1$ | $t{=}2$ | $t{=}3$ | $t{=}T$ |
| TrajectoryNet (Tong et al., 2020) | 0.62 | 1.15 | 1.49 | 1.26 | 0.99 |
| IPML (Vargas et al., 2021) | **0.34** | 1.13 | 1.35 | 1.01 | **0.49** |
| IPFP (no observations) | 0.57 | 1.53 | 1.86 | 1.32 | 0.85 |
| ISB (single-cell observations) | 0.57 | **1.04** | **1.24** | **0.94** | 0.83 |

access to snapshots of RNA data. This leads to a data set over 5 time ranges, the first from days 0–3 and the last from days 15–18. In the experiment, we followed the protocol by Tong et al. (2020), reduced the data dimensionality to $d = 5$ using PCA, and used the first and last time ranges as the initial and terminal constraints. All other time ranges are considered observational data. Contrary to the other experiments, intermediate data are imprecise (only a time range of multiple days is known) but abundant.

We learned the ISB using a zero drift and compared it against an unconditional bridge obtained through the IPFP (De Bortoli et al., 2021)—see Fig. 5. The ISB learns to generate trajectories

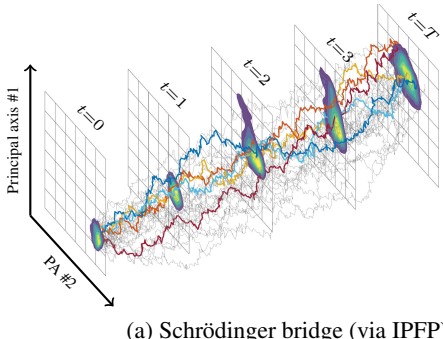

(a) Schrödinger bridge (via IPFP)

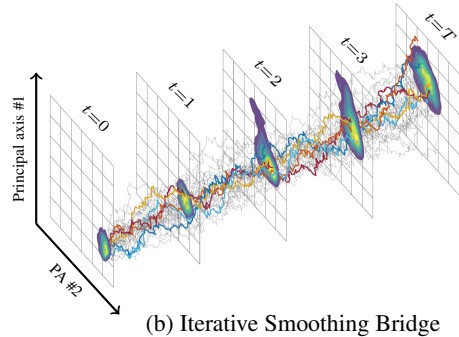

(b) Iterative Smoothing Bridge

Figure 5: Illustration of the trajectories of the high-dimensional single-cell experiment for the Schrödinger bridge (a) and the ISB (b), projected onto the first two principal components. The first five trajectories are highlighted in colour, and intermediate observation densities visualized as slices.

with marginals closer to the observed data while performing comparably to the IPFP at the initial and terminal stages. This improvement is also verified numerically in Table 2, showing that the ISB obtains a lower Earth mover's distance between the generated marginals and the observational data than IPFP. Additionally, Table 2 lists the performance of previous works that do not use the intermediate data during training (Tong et al., 2020) or only use it to construct an informative reference drift (Vargas et al., 2021), see App. B.5 for details. In both cases, ISB outperforms the other approaches w.r.t. the intermediate marginal distributions ($t = 1, 2, 3$), while IPML (Vargas et al., 2021) outperforms ISB at the initial and terminal stages due to its data-driven reference drift. Notice that while we reduced the dimensionality via PCA to $5$ for fair comparisons to Vargas et al. (2021), the ISB model would also allow modelling the full state-space model, with observations in the high-dimensional gene space and a latent SDE.

## 5 DISCUSSION AND CONCLUSION

The dynamic Schrödinger bridge problem provides an appealing setting for posing optimal transport problems as learning non-linear diffusion processes and enables efficient iterative solvers. However, while recent works have state-of-the-art performance in many complex application domains, they are typically limited to learning bridges with only initial and terminal constraints dependent on observed data. In this work, we have extended this paradigm and introduced the Iterative Smoothing Bridge (ISB), an iterative algorithm for learning data-conditional smoothing bridges. For this, we leveraged the strong connections between the constrained bridging problem and particle filtering in sequential Monte Carlo, extending them from pure inference to learning. We thoroughly assessed the applicability and flexibility of our approach in various experimental settings, including synthetic data sets and complex real-world scenarios (*e.g.*, bird migration, conditional image generation, and modelling single-cell RNA-sequencing time-series). In our experiments, we showed that ISB generalizes well to high-dimensional data, is computationally efficient, and provides accurate estimates of the marginals at intermediate and terminal times.

Accurately modelling the dynamics of complex systems under both path constraints induced by sparse observations and initial and terminal constraints is a key challenge in many application domains. These include biomedical applications, demographic modelling, and environmental dynamics, but also machine learning specific applications such as reinforcement learning, planning, and time-series modelling. All these applications have in common that the dynamic nature of the problem is driven by progression of time, and not only progression of a generative process as often is the case in, *e.g.*, generative image models. Thus, constraints over intermediate stages have a natural role and interpretation in this wider set of dynamic diffusion modelling applications. We believe the proposed ISB algorithm opens up new avenues for diffusion models in relevant real-world modelling tasks and will be stimulating for future work. For example, more sophisticated observational models, alternative strategies to account for multiple observations, and different noise schedules could be explored. Furthermore, the proposed approach could be extended to other types of optimal transport problems, such as the Wasserstein barycenter, a frequently employed case of the multi-marginal optimal transport problem.

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

# A  METHOD DETAILS

We present the details of the objective function derivation in App. A.1 and explain the connection of the backward drift function to Hamilton–Jacobi equations in App. A.2. In App. A.3, we discuss the behaviour of our model at the limit $M \to \infty$, that is, when the observations fully represent the marginal densities of the stochastic process.

## A.1  DERIVING THE MEAN-MATCHING LOSS AT OBSERVATION TIMES

**Proposition 1.** *Define the forward SDE as*

$$\mathrm{d}\mathbf{x}_t = f_{l,\theta}(\mathbf{x}_t, t)\,\mathrm{d}t + g(t)\,\mathrm{d}\boldsymbol{\beta}_t, \qquad\qquad \mathbf{x}_0 \sim \pi_0, \tag{12}$$

*and a backward SDE drift as*

$$b_{l,\phi}(\mathbf{x}_{t_{k+1}}, t_{k+1}) = f_{l-1,\theta}(\mathbf{x}_{t_{k+1}}, t_k) - g(t_{k+1})^2 \nabla \ln p_{t_{k+1}}, \tag{13}$$

*where $p_{t_{k+1}}$ is the particle filtering density after differential resampling at time $t_{k+1}$. Then $b_{l,\phi}(\mathbf{x}_{t_{k+1}}, t_{k+1})$ minimizes the loss function*

$$\ell^i_{k+1,obs} = \| b_{l,\phi}(\mathbf{x}^i_{t_{k+1}}, t_{k+1})\Delta_k - \mathbf{x}^i_{t_{k+1}} - f_{l-1,\theta}(\mathbf{x}^i_{t_{k+1}}, t_k)\Delta_k$$
$$+ \frac{1}{C_{\varepsilon,i}} \sum_{n=1}^N T_{(\varepsilon),i,n}\left(\mathbf{x}^n_{t_k} + f_{l-1,\theta}(\mathbf{x}^n_{t_k}, t_k)\Delta_k\right) \|^2, \tag{14}$$

*where we denote $\mathbf{C}_{\varepsilon,i} = \frac{1}{g(t_{k+1})^2\Delta_k} \mathrm{Var}\left(\sum_{n=1}^N T_{(\varepsilon),i,n}\tilde{\mathbf{x}}^n_{t_{k+1}}\right)$, and $\{\tilde{\mathbf{x}}^i_{t_{k+1}}\}_{i=1}^N$ are the particles before resampling.*

**Proof sketch.** Our objective is to find a backward drift function $b_{l,\phi}(\mathbf{x}_{t_{k+1}}, t_{k+1})$ as in Eq. (13). Notice that at observation times $t_k$, this is not equivalent to finding the reverse drift of the SDE forward transition and differential resampling combined, since the drift function $f_{l-1,\theta}$ alone does not map the particles $\{\mathbf{x}^i_{t_k}\}_{i=1}^N$ to the particles $\{\mathbf{x}^i_{t_{k+1}}\}_{i=1}^N$. We will derive a loss function for learning the backward drift as in Eq. (13) below, leaving the discussion on why it is a meaningful choice of a backward drift to App. A.2. Our derivation closely follows the proof of Proposition 3 in De Bortoli et al. (2021), but we provide the details here for the sake of completeness.

First, we give the transition density $p_{\mathbf{x}_{t_k} \mid \mathbf{x}^i_{t_{k-1}}}(\mathbf{x}_k)$ and apply it to derive the observation time loss $\ell^i_{k,\text{obs}}$. The derivation for the loss $\ell^i_{k,\text{no obs}}$ is skipped, since it is as in proof of Proposition 3 in De Bortoli et al. (2021). Suppose that at $t_k$, there are observations. By definition, the particles before resampling $\{\tilde{\mathbf{x}}^i_{t_{k+1}}\}_{i=1}^N$ are generated by the Gaussian transition density

$$p(\tilde{\mathbf{x}}_{t_{k+1}} \mid \mathbf{x}^i_{t_k}) = \mathrm{N}(\tilde{\mathbf{x}}_{t_{k+1}} \mid \mathbf{x}^i_{t_k} + \delta_k f_l(\mathbf{x}^i_{t_k}, t_k), g(t_{k+1})^2\Delta_k\mathbf{I}). \tag{15}$$

Recall that the resampled particles are defined as a weighted average of all the particles, $\mathbf{x}^i_{t_k} = \sum_{n=1}^N \tilde{\mathbf{x}}^n_{t_k} T_{(\varepsilon),i,n}$. Thus, the transition density from $\{\mathbf{x}^i_{t_k}\}_{i=1}^N$ to the particles $\{\mathbf{x}^i_{t_{k+1}}\}_{i=1}^N$ is also a Gaussian,

$$p(\mathbf{x}_{t^i_{k+1}} \mid \mathbf{x}^i_{t_k}) = \mathrm{N}(\tilde{\mathbf{x}}_{t_{k+1}} \mid \sum_{n=1}^N T_{(\varepsilon),i,n}(\mathbf{x}^n_{t_{k-1}} + \Delta_k f_{l-1,\theta}(\mathbf{x}^n_{t_k}, t_k)), g(t_{k+1})^2\Delta_k C_{\varepsilon,i}\mathbf{I}_d). \tag{16}$$

We will derive the loss function Eq. (9) by modifying the mean matching proof in De Bortoli et al. (2021) by the transition mean Eq. (16) and the backward drift definition Eq. (13). Using the particle filtering approximation, the marginal density can be decomposed as $p_{t_{k+1}}(\mathbf{x}_{k+1}) = \sum_{i=1}^N p_{t_k}(\mathbf{x}^i_k)p_{\mathbf{x}_{k+1} \mid \mathbf{x}^i_k}(\mathbf{x}_{k+1})$. By substituting the transition density Eq. (16) it follows that

$$p_{t_{k+1}}(\mathbf{x}_{t_{k+1}})$$
$$= \frac{1}{Z} \sum_{i=1}^N p_{t_k}(\mathbf{x}^i_{t_k}) \exp\left(-\frac{\|\left(\sum_{n=1}^N T_{(\varepsilon),i,n}(\mathbf{x}^i_{t_k} + f_{l-1,\theta}(\mathbf{x}_{t_k}, t_k))\right) - \mathbf{x}_{t_{k+1}}\|^2}{2g(t_{k+1})^2 C_{\varepsilon,i}\Delta_k}\right), \tag{17}$$

where $Z$ is the normalization constant of Eq. (16). As in the proof of Proposition 3 of De Bortoli et al. (2021), we derive an expression for the score function. Since $\nabla \ln p_{t_{k+1}}(\mathbf{x}_{t_{k+1}}) = \frac{\nabla_{\mathbf{x}_{t_{k+1}}} p_{t_{k+1}}(\mathbf{x}_{t_{k+1}})}{p_{t_{k+1}}(\mathbf{x}_{t_{k+1}})}$, we first manipulate $\nabla_{\mathbf{x}_{t_{k+1}}} p_{t_{k+1}}(\mathbf{x}_{t_{k+1}})$,

$$\nabla_{\mathbf{x}_{t_{k+1}}} p_{t_{k+1}}(\mathbf{x}_{t_{k+1}}) \tag{18}$$

$$= \frac{1}{Z} \sum_{i=1}^{N} \nabla_{\mathbf{x}_{t_{k+1}}} p(\mathbf{x}_{t_k}^i) \exp\left( -\frac{\left\| \left( \sum_{n=1}^{N} T_{(\varepsilon),i,n}(\mathbf{x}_{t_k}^i + f_{l-1,\theta}(\mathbf{x}_{t_k}, t_k)) \right) - \mathbf{x}_{t_{k+1}} \right\|^2}{2g(t_{k+1})^2 C_{\varepsilon,i}\Delta_k} \right) \tag{19}$$

$$= \frac{1}{Z} \left( \sum_{i=1}^{N} p(\mathbf{x}_{t_k}^i) \left( \sum_{n=1}^{N} \frac{1}{g(t_{k+1})^2 \Delta_k C_{\varepsilon,i}} \left( T_{(\varepsilon),i,n}(\mathbf{x}_{t_k}^i + f_{l-1,\theta}(\mathbf{x}_{t_k}, t_k)) - \mathbf{x}_{t_{k+1}} \right) \right) \right. \tag{20}$$

$$\exp\left( -\frac{\left\| \left( \sum_{n=1}^{N} T_{(\varepsilon),i,n}(\mathbf{x}_{t_k}^i + f_{l-1,\theta}(\mathbf{x}_{t_k}, t_k)) \right) - \mathbf{x}_{t_{k+1}} \right\|^2}{2g(t_{k+1})^2 C_{\varepsilon,i}\Delta_k} \right) \right). \tag{21}$$

Substituting $p_{t_k}(x_k^i) = \frac{p_{t_{k+1}}(\mathbf{x}_{t_{k+1}}) p_{\mathbf{x}_{k+1} \mid \mathbf{x}_k^i}(\mathbf{x}_{k+1})}{p_{\mathbf{x}_k^i \mid \mathbf{x}_{k+1}}(\mathbf{x}_k^i)}$ to the equation above gives

$$\nabla_{\mathbf{x}_{t_{k+1}}} p_{t_{k+1}}(\mathbf{x}_{t_{k+1}})$$

$$= p_{t_{k+1}}(\mathbf{x}_{t_{k+1}}) \sum_{i=1}^{N} p_{\mathbf{x}_{k+1} \mid \mathbf{x}_k^i}(\mathbf{x}_k^i) \left( \sum_{n=1}^{N} \frac{\left( T_{(\varepsilon),i,n}(\mathbf{x}_{t_k}^i + f_{l-1,\theta}(\mathbf{x}_{t_k}, t_k)) - \mathbf{x}_{t_{k+1}} \right)}{g(t_{k+1})^2 \Delta_k C_{\varepsilon,i}} \right), \tag{22}$$

and dividing by $p_{t_{k+1}}(\mathbf{x}_{t_{k+1}})$ yields

$$\nabla \ln p_{t_{k+1}}(\mathbf{x}_{t_{k+1}})$$

$$= \sum_{i=1}^{N} p_{\mathbf{x}_{t_k^i} \mid \mathbf{x}_{t_{k+1}}}(\mathbf{x}_{t_k^i}) \left( \sum_{n=1}^{N} \frac{\left( T_{(\varepsilon),i,n}(\mathbf{x}_{t_k}^i + f_{l-1,\theta}(\mathbf{x}_{t_k}, t_k)) - \mathbf{x}_{t_{k+1}} \right)}{g(t_{k+1})^2 \Delta_k C_{\varepsilon,i}} \right). \tag{23}$$

Substituting Eq. (23) to the definition of the optimal backward drift Eq. (13) gives

$$b_{l,\phi}(\mathbf{x}_{t_{k+1}}, t_{k+1})$$
$$= f_{l-1,\theta}(\mathbf{x}_{t_{k+1}}, t_k) - g(t_{k+1})^2 \nabla \ln p_{t_{k+1}}(\mathbf{x}_{k+1})$$
$$= f_{l-1,\theta}(\mathbf{x}_{t_{k+1}}, t_k) - \tag{24}$$
$$g(t_{k+1})^2 \sum_{i=1}^{N} p_{\mathbf{x}_{t_k^i} \mid \mathbf{x}_{t_{k+1}}}(\mathbf{x}_{t_{k+1}}) \left( \sum_{n=1}^{N} \frac{\left( T_{(\varepsilon),i,n}(\mathbf{x}_{t_k}^i + f_{l-1,\theta}(\mathbf{x}_{t_k}, t_k)) - \mathbf{x}_{t_{k+1}} \right)}{g(t_{k+1})^2 \Delta_k C_{\varepsilon,i}} \right),$$

where taking $f_{l-1,\theta}(\mathbf{x}_{t_{k+1}}, t_k)$ inside the sum yields

$$b_{l,\phi}(\mathbf{x}_{t_{k+1}}, t_{k+1}) = \left( \sum_{i=1}^{N} p_{\mathbf{x}_{t_k^i} \mid \mathbf{x}_{t_{k+1}}}(\mathbf{x}_{t_{k+1}}) \right.$$
$$\left( \frac{1}{C_{\varepsilon,i}} \left( \sum_{n=1}^{N} T_{(\varepsilon),i,n}(\mathbf{x}_{t_k}^i + f_{l-1,\theta}(\mathbf{x}_{t_k}, t_k)) \right) - \frac{\mathbf{x}_{t_{k+1}}}{C_{\varepsilon,i}} - \Delta_k f_{l-1,\theta}(\mathbf{x}_{t_{k+1}}, t_k) \right) / \Delta_k \right). \tag{25}$$

Multiplying the equation above by $\Delta_k$ gives

$$\Delta_k b_{l,\phi}(\mathbf{x}_{t_{k+1}}^i, t_{k+1})$$
$$= \left( \sum_{n=1}^{N} T_{(\varepsilon),i,n}(\mathbf{x}_{t_k}^n + f_{l-1,\theta}(\mathbf{x}_{t_k}^n, t_k)) \right) - \frac{\mathbf{x}_{t_{k+1}}^i}{C_{\varepsilon,i}} - \Delta_k f_{l-1,\theta}(\mathbf{x}_{t_{k+1}^i}, t_k). \tag{26}$$

Thus we may set the objective for finding the optimal backward drift $b_{l,\phi}$ as

$$\ell_{k+1,\text{no obs}}^i = \| b_{l,\phi}(\mathbf{x}_{t_{k+1}}^i, t_{k+1})\Delta_k - \frac{\mathbf{x}_{t_{k+1}}^i}{C_{\varepsilon,i}} - f_{l-1,\theta}(\mathbf{x}_{t_{k+1}}^i, t_k)\Delta_k$$
$$+ \frac{1}{C_{\varepsilon,i}} \sum_{n=1}^{N} T_{(\varepsilon),i,n} \left( \mathbf{x}_{t_k}^n + f_{l-1,\theta}(\mathbf{x}_{t_k}^n, t_k)\Delta_k \right) \|^2. \tag{27}$$

□

Notice that if the weights before resampling are uniform, then $T_{(\varepsilon)} = \mathbf{I}_N$, and for all $i \in 1, 2, \ldots, N$ it holds that $C_{\varepsilon,i} = 1$, since all but one of the terms in the sum $\frac{1}{g(t_{k+1})^2} \mathrm{Var} \left( \sum_{n=1}^N T_{(\varepsilon),i,n} \tilde{\mathbf{x}}_{t_{k+1}}^n \right)$ vanish. Similarly, for one-hot weights $C_{\varepsilon,i} = 1$. In practice, we set the constant $C_{\varepsilon,i} = 1$ as in Eq. (9) and observe good empirical performance with the simplified loss function.

## A.2 CONNECTION TO HAMILTON–JACOBI EQUATIONS

We connect the backward drift function $b_{l,\phi}(\mathbf{x}_{t_{k+1}}, t_{k+1}) = f_{l-1,\theta}(\mathbf{x}_{t_{k+1}}, t_k) - g(t_{k+1})^2 \nabla \ln p_{t_{k+1}}(\mathbf{x}_{t_{k+1}})$ to the Hamilton–Jacobi equations for stochastic control through following the setting of Maoutsa & Opper (2021), which applies the drift $f_{l-1,\theta}(\mathbf{x}_t, t) - g(t)^2 \nabla \ln p_t(\mathbf{x}_t)$ for a backwards SDE initialized at $\pi_T$.

Consider a stochastic control problem with a path constraint $U(\mathbf{x}_t, t)$, optimizing the following loss function,

$$\mathcal{J} = \frac{1}{N} \sum_{i=1}^N \int_{t=0}^T \frac{1}{2g(t)^2} \| f_\theta(\mathbf{x}_t^i, t) - f(\mathbf{x}_t^i, t) \|^2 + U(\mathbf{x}_t^i, t) \, \mathrm{d}t - \ln \chi(\mathbf{x}_T^i), \tag{28}$$

with the paths $\mathbf{x}_t^i$ sampled as trajectories from the SDE

$$\mathbf{x}_0 \sim \pi_0, \quad \mathrm{d}\mathbf{x}_t = f_{l-1,\theta}(\mathbf{x}_t, t) \, \mathrm{d}t + g(t) \, \mathrm{d}\boldsymbol{\beta}_t, \tag{29}$$

and the loss $\ln \chi(\mathbf{x}_T^i)$ measures distance from the distribution $\pi_T$. Since we set the path constraint via observational data, our method resembles setting $U(\mathbf{x}_t^i, t) = 0$ when $t$ is not an observation time, and $U(\mathbf{x}_t^i) = -\log \mathbf{p}(\mathbf{y} \mid \mathbf{x}_t^i)$, where $\mathbf{p}(\mathbf{y} \mid \mathbf{x}_t^i)$ is the observation model.

Let $q_t(\mathbf{x})$ denote the marginal density of the controlled (drift $f_\theta$) SDE at time $t$. In Maoutsa & Opper (2021), the marginal density is decomposed as

$$q_t(\mathbf{x}) = \varphi_t(\mathbf{x}) p_t(\mathbf{x}), \tag{30}$$

where $\varphi_t(\mathbf{x})$ is a solution to a backwards Fokker-Planck-Kolmogorov (FPK) partial differential equation starting from $\varphi_T(\mathbf{x}) = \pi_T$, and the density evolves as in

$$\frac{\mathrm{d}\varphi_t(\mathbf{x})}{\mathrm{d}t} = -\mathcal{L}_f^\dagger \varphi_t(\mathbf{x}) + U(\mathbf{x}, t)\varphi_t(\mathbf{x}), \tag{31}$$

where $\mathcal{L}_f^\dagger$ is the adjoint FPK operator to the uncontrolled system. The density $p_t(\mathbf{x})$ corresponds to the forward filtering problem, initialized with $\pi_0$,

$$\frac{\mathrm{d}p_t(\mathbf{x})}{\mathrm{d}t} = \mathcal{L}_f(p_t(\mathbf{x})) - U(\mathbf{x}, t)p_t(\mathbf{x}), \tag{32}$$

where $\mathcal{L}_f$ is the FPK operator of the uncontrolled SDE (with drift $f$). The particle filtering trajectories $\{\mathbf{x}_{t_k}\}^i$ generated in our method are samples from the density defined by Eq. (32). In the context of our method, the path constraint matches the log-weights of particle filtering at observation times and is zero elsewhere.

In Maoutsa & Opper (2021), a backward evolution for $q_t$ is applied, using the backwards time $\tilde{q}_{T-\tau}(\mathbf{x}) = q_\tau(\mathbf{x})$, yielding a backwards SDE starting from $\tilde{q}_0(\mathbf{x}) = \{\mathbf{x}_T^i\}_{i=1}^N$, reweighted according to $\pi_T$. The backward samples from $\tilde{q}$ are generated following the SDE dynamics

$$\mathrm{d}\mathbf{x}_\tau^i = (f(\mathbf{x}_\tau^i, T - \tau) + g(t)^2 \nabla \ln p_{T-\tau}(\mathbf{x}_\tau^i) \, \mathrm{d}t + g(t) \, \mathrm{d}\beta_\tau. \tag{33}$$

We have thus selected the backward drift $b_{l,\phi}$ to match the drift of $\tilde{q}_t(x)$, the backward controlled density. Intuitively, our choice of $b_{l,\phi}$ is a drift which generates the smoothed particles when initialized at $\{\mathbf{x}_T^i\}_{i=1}^N$, the terminal state of the forward SDE. The discrepancy between $\pi_T$ and the distribution induced by $\{\mathbf{x}_T^i\}_{i=1}^N$ then motivates the use of an iterative scheme after learning to simulate from $q_t(x)$.

A.3   OBSERVING THE FULL MARGINAL DENSITY

Suppose that at time $t_k$, we let the number of observations grow unbounded. We analyse the behaviour of our model at the resampling step, at the limit $M \to \infty$ for the number of observations and $\sigma \to 0$ for the observation noise. When applying the bootstrap proposal, recall that we combined the multiple observations to compute the log-weights as

$$\log w_{t_k}^i = -\frac{1}{2\sigma^2} \sum_{\mathbf{y}_j \in \mathcal{D}_{i,t_k}^H} \|\mathbf{x}_{t_k}^i - \mathbf{y}_j\|^2, \tag{34}$$

which works well in practice for the sparse-data settings we have considered. Below we analyse the behaviour of an alternative way to combine the weights and show that given an infinite number of observations, it creates samples from the true underlying distribution.

**Proposition 2.** *Let $\{\mathbf{x}_{t_k}^i\}_{i=1}^N$ be a set of particles and $\{\mathbf{y}_j\}_{j=1}^M$ the observations at time $t_k$. Assume that the observations have been sampled from a density $\rho_{t_k}$ and that for all $i$ it holds that $\mathbf{x}_{t_k}^i \in supp(\rho_{t_k})$. Define the particle weights as*

$$\log w_{t_k,\sigma,M}^i = \log \left( \frac{1}{Z|\mathcal{D}_{i,t_k}^{H(M)}|} \sum_{\mathbf{y}_j \in \mathcal{D}_{i,t_k}^{H(M)}} \exp(-\|\mathbf{x}_{t_k}^i - \mathbf{y}_j\|^2/2\sigma^2) \right), \tag{35}$$

*where $Z$ is the normalization constant of the observation model Gaussian $p(\mathbf{y} \mid \mathbf{x}_{t_k}^i)$. Then for each particle $\mathbf{x}_{t_k}^i$, its weight satisfies*

$$\lim_{\sigma \to 0} \lim_{M \to \infty} w_{t_k,\sigma,M}^i = \rho_{t_k}(x_{t_k}) \tag{36}$$

**Proof sketch.** We drop the $\sigma$ and $H(M)$ from the weight notation for simplicity of notation, but remark that the particle filtering weights are dependent of both quantities. Consider the number of particles $N$ fixed, and denote the $d$-dimensional sphere centered at $\mathbf{x}_{t_k}^i$ as $B(\mathbf{x}_{t_k}^i, r)$. Since each particle $\mathbf{x}_{t_k}^i$ lies in the support of the true underlying marginal density $\rho_{t_k}$, then for any radius $r > 0$ such that $B(\mathbf{x}_{t_k}^i, r) \in supp(\rho_{t_k})$, and $H > 0$, we may choose $M$ high enough so that the points $\mathbf{y}_j \in \mathcal{D}_{i,t_k}^H$ satisfy $\mathbf{y}_j \in B(\mathbf{x}_{t_k}^i, r)$. It follows from Eq. (35) that

$$w_{t_k}^i = \frac{1}{Z|\mathcal{D}_{i,t_k}^{H(M)}|} \sum_{\mathbf{y}_j \in \mathcal{D}_{i,t_k}^{H(M)}} \exp(-\|\mathbf{x}_{t_k}^i - \mathbf{y}_j\|^2/2\sigma^2). \tag{37}$$

For any $r > 0$ and with observation noise $\sigma = cr$, we may set $c, H(M)$ so that the sum above approximates the integral

$$w_{r,t_k}^i \approx \frac{1}{|B(\mathbf{x}_{t_k}^i, r)|} \int_{B(\mathbf{x}_{t_k}^i, r)} p(\mathbf{y} \mid \mathbf{x}_{t_k}^i)\rho_t(\mathbf{y}) \, d\mathbf{y}. \tag{38}$$

By applying the Lebesque differentiation theorem, we obtain that for almost every $\mathbf{x}_{t_k}^i$, we have $\lim_{r \to 0} w_{t_k,r}^i = \rho_{t_k}(\mathbf{x}_{t_k}^i)$, since as $\sigma \to 0$, the density $p(\mathbf{y} \mid \mathbf{x}_{t_k}^i)$ collapses to the Dirac delta of $\mathbf{x}_{t_k}^i$. □

Prop. 2 can be interpreted as the infinite limit of a kernel density estimate of the true underlying distribution. Resampling accurately reweights the particles so that the probability of resampling particle $\mathbf{x}_{t_k}^i$ is proportional to the density $\rho_{t_k}$ compared to the other particles. Notice that the result does not guarantee that the particles will cover the support of $\rho_{t_k}$, since we did not assume that the drift initialization generates a marginal density at time $t_k$ covering its support.

B   EXPERIMENTAL DETAILS

B.1   2D TOY DATA SETS

For the constrained transport problem for two-dimensional scikit-learn, the observational data we chose to use was different for each of the three data sets presented; two moons, two circles and the

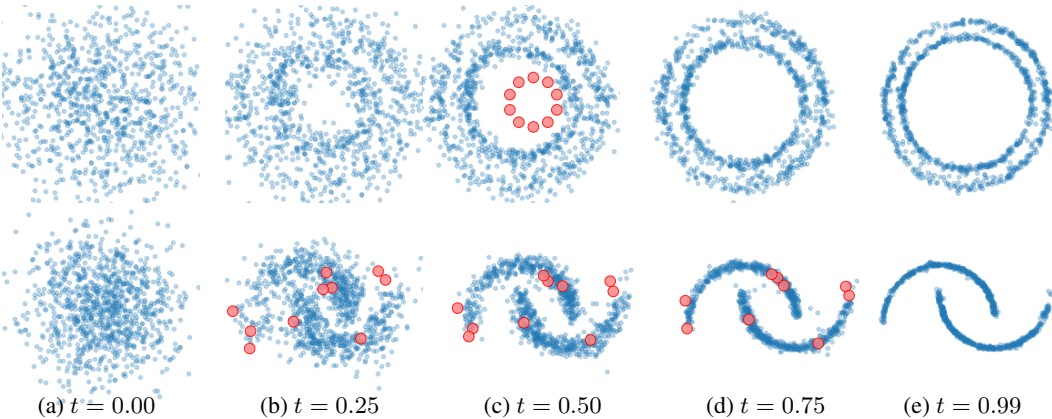

(a) $t = 0.00$      (b) $t = 0.25$      (c) $t = 0.50$      (d) $t = 0.75$      (e) $t = 0.99$

Figure 6: The IPFP result for the experiment in Fig. 3 in the main paper.2D toy experiments, where observations (red markers) not used while training but included in the figure for reference. The dynamics learned by IPFP are clearly different from the ISB learned dynamics.

S-shape. All three experiments had the same discretization ($t \in [0, 0.99]$), $\Delta_k = 0.01$), learning rate 0.001, and differentiable resampling regularization parameter $\varepsilon = 0.01$. The process noise $g(t)^2$ follows a linear schedule from 0.001 to 1, with low noise at time $t = 0$ and high noise at $t = 0.99$, and each iteration of the ISB method trains the forward and backward drift networks each for 5000 iterations, with batch size 256. Other hyperparameters are explained below.

**Two moons** The observational data consists of 10 points selected from the Schrödinger bridge trajectories, all observed at $t \in [0.25, 0.5, 0.75]$ with an exponential observation noise schedule $\kappa(l) = 1.25^{l-1}$. The ISB was ran for 6 epochs, and initialized with a drift from the pre-trained Schrödinger bridge model from the unconstrained problem.

**Two circles** The observational data consists of 10 points which lie evenly distributed on a circle, observed at $t = 0.5$ with an exponential observational noise schedule $\kappa(l) = 0.5 \cdot 1.25^{l-1}$. The ISB was ran for 6 epochs, and initialized with a drift from the pre-trained Schrödinger bridge model from the unconstrained problem.

**S-shape** The observational data consists of 6 points, with pairs being observed at times $t \in [0.4, 0.5, 0.6]$. We used a bilinear observational noise schedule with a linear decay for the first half of the iterations from $\kappa(0)^2 = 4$ to $\kappa(L/2)^2 = 1$ and a linear ascend for the second half of the iterations from $\kappa(L/2)^2 = 1$ to $\kappa(L)^2 = 4$. The ISB ran for 6 epochs, with a zero drift initialization.

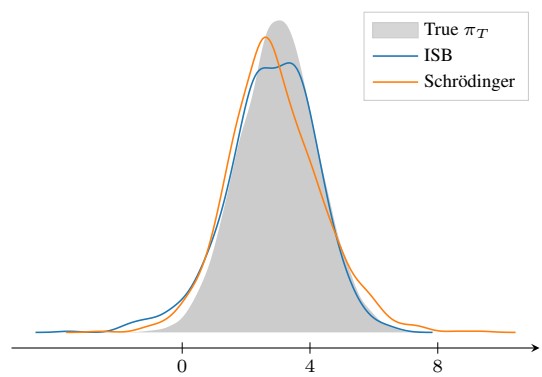

Figure 7: A kernel density estimate of the Beneš SDE terminal state. We compare $\pi_T$ to the Schrödinger bridge and ISB terminal states. Both unconstrained Schrödinger bridge and ISB terminal states succeed in representing $\pi_T$ well, with the Schrödinger bridge terminal state more closely matching $\pi_T$ near its mean.

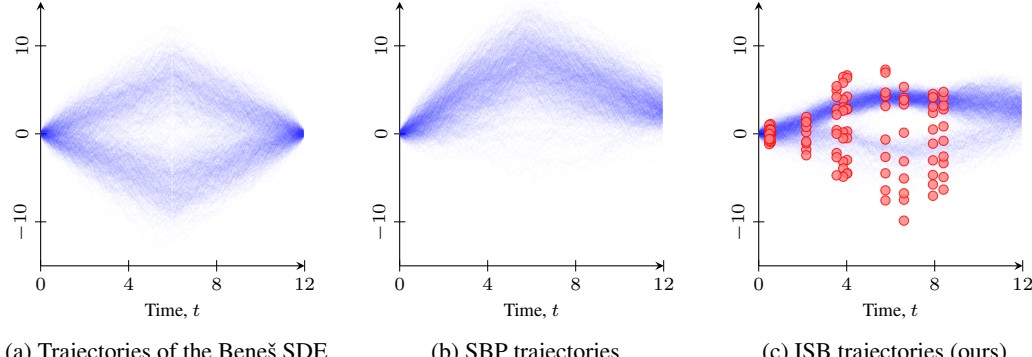

(a) Trajectories of the Beneš SDE      (b) SBP trajectories      (c) ISB trajectories (ours)

Figure 8: Comparison of the solution for the SBP (with Beneš SDE reference drift) and the ISB (with zero initial drift) on the Beneš SDE under sparse observations (●). The target distribution $\pi_T$ is slightly shifted and scaled from the Beneš SDE. Even if the SBP has the true model as reference drift, its trajectories degenerate into a unimodal distribution, while the ISB manages to cover both modes even if only sparse observations are available.

## B.2   THE BENEŠ SDE

In the Beneš SDE experiment, we obtain the sparse observational data from sampled Beneš SDE trajectories while the terminal state is a shifted and scaled $(3 + 5x_T)$ version of a Beneš marginal density. As the Beneš trajectories were first generated by simulating the SDE until $t = 6$ and then in reverse from $t = 6$ to $t = 0$, we set $T = 11.97$. We apply the analytical expression for the Beneš marginal density for computing $\log p_t(\mathbf{x})$,

$$p_t(\mathbf{x}) = \frac{1}{\sqrt{2\pi t}} \frac{\cosh(\mathbf{x})}{\cosh(\mathbf{x}_0)} \exp\left(-\frac{1}{2}t\right) \exp\left(-\frac{1}{2t}(\mathbf{x} - \mathbf{x}_0)^2\right). \tag{39}$$

See the Beneš SDE trajectories in Fig. 8a. As expected, the transport model with no observations performs well in the generative task, but its trajectories cover also some low-likelihood space around $t = 6$ (in the middle part in Fig. 8b). The observations for the ISB model were sampled from the generated trajectories, 10 observations at 10 random time-instances (see Fig. 8c)

Both the unconstrained Schrödinger bridge model and the ISB model were ran for 3 iterations, using a learning rate of 0.001 for the neural networks. Likely due to the fact that the problem was only one-dimensional, convergence of the Schrödinger bridge to a process which matches the desired terminal state was fast, and we chose not to run the model for a higher number of ISB iterations, see Fig. 7 for a comparison of the trained model marginal densities and the true terminal distribution $\pi_T$. We set the observation noise schedule to the constant 0.7, and at each iteration of the ISB or the unconstrained Schrödinger bridge the drift neural networks were trained for 5000 iterations each with the batch size 256, and the trajectories were refreshed every 500 iterations with a cache size of 1000 particles. The number of nearest neighbours to compare to was $H = 10$.

## B.3   THE BIRD MIGRATION DATA SET

The ISB model learned bird migration trajectories which transport the particles from the Northern Europe summer habitats to the southern winter habitats, see Fig. 10 for a comparison of a Schrödinger bridge and ISB. Since the problem lies on a sphere, Schrödinger bridge methods adjusted for learning on Riemannian manifolds could have been applied here. For simplicity we mapped the probelm to a two-dimensional plane using a Mercator projection, and solved the problem on a $[0, 5] \times [0, 5]$ square. The SDE had the discretization $t \in [0, 0.99]$, $\Delta_k = 0.01$ and a constant process noise $g(t)^2 = 0.05$. The model was trained for 12 iterations, and initialized with a zero drift, while the observational data was chosen by the authors to promote learning trajectories clearly different from the unconstrained transport trajectories. The observation noise schedule was piecewise linear (starting at 2, going to 0.1 at iteration 6, then rising linearly to reach 2 at iteration 12). At each ISB iteration, the neural networks were trained for 5000 iterations each, and the trajectories refreshed every 1000 iterations. We used a batch size of 256 and learning rate 0.001.

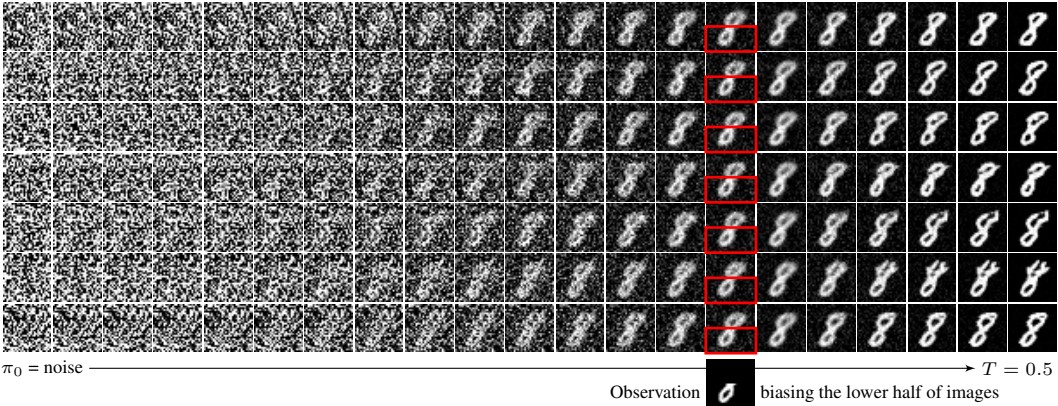

$\pi_0$ = noise $\longrightarrow$ $T = 0.5$

Observation    biasing the lower half of images

Figure 9: Model trajectories for MNIST digit '8' conditioned on a lower-loop of a single '8' at $t = 0.38$ to bias the lower half of the digits to look alike, with the effect still visible at terminal time $T$.

### B.4 THE MNIST GENERATION TASK

Applying state-space model approaches such as particle filtering and smoothing to generative diffusion models directly in the observation space (that is, not in a lower-dimensional latent space) has to our knowledge not been explored before. Some experimental design choices had a great impact into the training objectives sensibility, as the observational data is completely artificial and its timing during the process modifies the filtering distribution significantly. As the MNIST conditional generative model was trained to display the scalability of our method beyond low-dimensional toy examples, we did not further explore optimizing the hyperparameters or the observation model. To avoid the background noise in MNIST images in the middle of the generative process impacting the particle filtering weights excessively, the observation model is a Gaussian with masked inputs equal to zero in pixels where the observation image is black, see Fig. 9 for sampled trajectories. The figure shows the progression of seven samples, where the lower half of the eights resemble the observation target.

The SDE was run for time $t \in [0, 0.5]$, with the digit eight observed at $t = 0.38$. The ISB method was applied for 10 iterations, with a discretization $t \in [0, 0.495]$, $\Delta_k = 0.005$, and the process noise $g(t)^2$ followed a linear schedule from 0.0001 to 1. At each iteration of the method, the forward and backward drift neural networks were trained for 5000 iterations with a batch size of 256, and the trajectory cache regenerated every 1000 iterations. The observational data consisted of a single sample of a lower half of the digit eight, observed at time $t = 0.38$. The observation noise schedule was a constant $\kappa(l) = 0.3$.

### B.5 SINGLE-CELL DATA SET

We directly use the preprocessed data from the TrajectoryNet (Tong et al., 2020) repository. A major difference between our implementation and Vargas et al. (2021) is the reference drift. We set the reference drift to zero, which means that we utilize the intermediate data only as observations in the state-space model. On the contrary, Vargas et al. (2021) fits a mixture model of 15 Gaussians on the combined data set (across all measurement times) and sets the reference drift to the gradient of the log likelihood of the mixture model. Effectively, such a reference drift aids in keeping the SDE trajectories within the support of the combined data set. We remark that if the intermediate observed marginals had clearly disjoint support, combining all the data would cause the mixture model to have 'gaps' and could cause an unstable reference model drift. Thus we consider our approach of setting the reference drift to zero as more generally applicable.

As in Vargas et al. (2021), we set the process noise to $g(t) = 1$ and model the SDE between time $t \in [0, 4]$. The learning rate is set to 0.001 with batch size 256 and number of neural network training iterations 5000, and we apply the ISB for 6 iterations. We filter using 1000 points from the intermediate data sets, but compute the Earth mover's distance by a comparison to all available data. As the observational data at $T = 1, 2, 3$ consists of a high number of data points, the parameters $H$ (number of nearest neighbours) and $\sigma$ (observation noise) need to be carefully set. We set $H = 10$

to only include the close neighbourhood of each particle, and set the observation noise schedule as constant 0.7.

## C  COMPUTATIONAL CONSIDERATIONS

In Sec. 3.2, we raised a number of important computational considerations for the constrained transport problem. Below we discuss them in detail, analyzing the limit $L \to \infty$ from the perspective of setting the observation noise schedule in App. C.1, and presenting ablation results on modifying the initial drift in the bird migration experiment in App. C.2.

### C.1  DISCUSSION ON OBSERVATION NOISE

We briefly mentioned in Sec. 3.2 that when letting $L \to \infty$, the choice of observation noise should be carefully planned in order for the ISB procedure to have a stationary point. Here we explain why an unbounded observation noise schedule $\kappa(l)$ implies convergence to the IPF method for uncontrolled Schrödinger bridges (De Bortoli et al., 2021), when using a nearest neighbour bootstrap filter as the proposal density.

**Proposition 3.** *Let $\Omega \in \mathbb{R}^d$ be a bounded domain where both the observations and SDE trajectories lie, and let the particle filtering weights $\{w_{l,t_k}^i\}_{i=1}^N$ be as in Eq. (11), but after normalization. If the schedule $\kappa(l)$ is unbounded with respect to $l$, then for any $\delta$ there exists $l'$ such that for the normalized weights it holds*

$$|\hat{w}_{l',t_k}^i - \frac{1}{N}| \le \delta. \tag{40}$$

**Proof sketch.** Since we set the proposal density to be the bootstrap filter, the observation weights at ISB iteration $l$ are equal to

$$\log w_{l,t_k}^i = -\frac{1}{2\kappa(l)^2} \sum_{\mathbf{y}_j \in \mathcal{D}_{t_k}^H} \|\mathbf{x}_{t_k}^i - \mathbf{y}_j\|^2. \tag{41}$$

Since $\kappa(l)$ is unbounded, for any $S > 0 \ \exists \ l'$ such that $\kappa(l') \ge S$. We choose the value of $S$ so that the following derivation yields Eq. (40).

Let $S = \sqrt{0.5 R^{-1} |\mathcal{D}_{t_k}^H| \operatorname{diam}(\Omega)^2}$, and apply the property that $\|\mathbf{x}_{t_k}^i - \mathbf{y}_j\|^2 \le \operatorname{diam}(\Omega)^2$ to Eq. (41),

$$
\begin{aligned}
\log w_{l',t_k}^i &\ge -\frac{1}{2S^2} \sum_{\mathbf{y}_j \in \mathcal{D}_{t_k}^H} \|\mathbf{x}_{t_k}^i - \mathbf{y}_j\|^2 \\
&\ge -\frac{\sum_{\mathbf{y}_j \in \mathcal{D}_{t_k}^H} \|\mathbf{x}_{t_k}^i - \mathbf{y}_j\|^2}{R^{-1} |\mathcal{D}_{t_k}^H| \operatorname{diam}(\Omega)^2} \ge -\frac{\sum_{\mathbf{y}_j \in \mathcal{D}_{t_k}^H} \operatorname{diam}(\Omega)^2}{R^{-1} |\mathcal{D}_{t_k}^H| \operatorname{diam}(\Omega)^2} \ge -R.
\end{aligned}
\tag{42}
$$

The bound above is for the unnormalized weights, and the normalized log-weights are defined as

$$\log \hat{w}_{l',t_k}^i = \log w_{l',t_k}^i - \log \left( \sum_{j=1}^N \exp(\log w_{l',t_k}^j) \right), \tag{43}$$

where for the normalizing constant it holds that

$$\log \left( \sum_{j=1}^N \exp(\log w_{l',t_k}^j) \right) \le \log \left( \sum_{j=1}^N 1 \right) = \log(N), \tag{44}$$

since $w_{l',t_k}^j$ is the value of a probability density and thus always $w_{l',t_k}^j \le 1$. Combining Eq. (43), Eq. (42) and Eq. (44), it follows that

$$\log \hat{w}_{l',t_k}^i - (-\log(N) \ge -R, \tag{45}$$

where taking exponentials on both sides gives

$$\hat{w}_{l',t_k}^i - \frac{1}{N} \ge -(1 - \exp(-R))\frac{1}{N}. \tag{46}$$

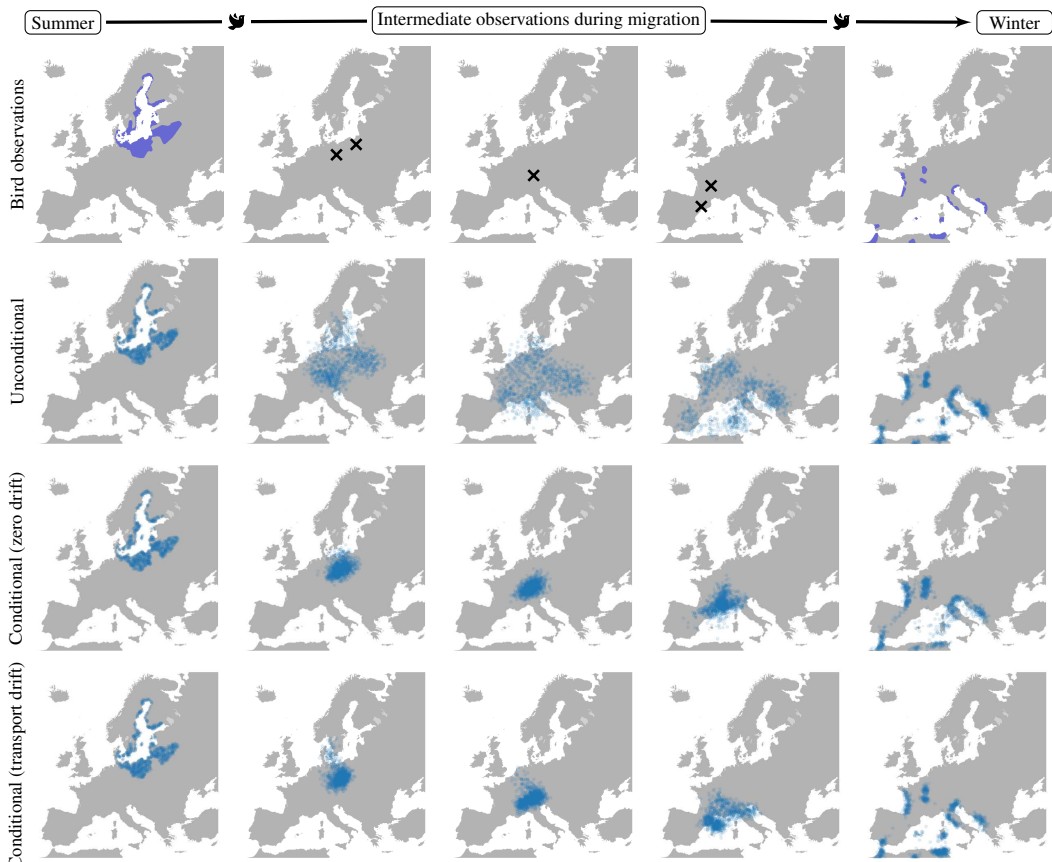

Figure 10: Top row: The first map image on the left describes the initial position of the birds, and the final on the right their position after migration. The observational data in the middle are bird observations during migration, at given timestamps. Second row: Marginal densities of a Schrödinger bridge model from the initial to terminal distribution, without using the observations. Third row: Marginal densities of our model, using both initial and terminal distributions and observational data and a zero drift initialization. Bottom row: Same as third row, but with the second row dynamics as initialization.

Since the weights are normalized, even the largest particle weight $\hat{w}^j_{l',t_k}$ can differ from $\frac{1}{N}$ as much as every smaller weight in total lies under $\frac{1}{N}$,

$$\hat{w}^j_{l',t_k} \leq \frac{1}{N} + (N-1)\left((1 - \exp(-R))\frac{1}{N}\right), \tag{47}$$

implying that for any weight $\hat{w}^j_{l',t_k}$, it holds that

$$|\hat{w}^j_{l',t_k} - \frac{1}{N}| \leq (N-1)\left((1 - \exp(-R))\frac{1}{N}\right) \leq 1 - \exp(-R), \tag{48}$$

and selecting $R = -\log(1 - \delta)$ is sufficient for $\delta < 1$. $\square$

Effectively, the above derivation implies that for an unbounded observation noise schedule $\kappa(l)$, the particle weights will converge to uniform weights. Since performing differentiable resampling on uniform weights implies that $\mathbf{T}_{(\varepsilon)} = \mathbf{I}_N$, the ISB method trajectory generation step and the objective in training Nthe backward drift converge to those of the IPF method for solving unconstrained Schrödinger bridges. Intuitively, this means that at the limit $L \to \infty$, our method will focus on reversing the trajectories and matching the terminal distribution while not further utilizing information from the observations.

## C.2 ABLATION ON INITIAL DRIFT

We conducted an ablation study on drift initialization for the bird migration problem. As the distributions $\pi_0$ and $\pi_T$ (as pictured in Fig. 10) are complex, we consider the problem setting to be interesting for setting $f_0$ as the unconstrained transport problem drift. To this end, we trained a Schrödinger bridge model for 10 epochs, and trained an ISB model with the same hyperparameter selections as explained in App. B.3, using the Schrödinger bridge as the initialization. Compare the two bottom rows of Fig. 10 to see a selection of marginal densities of the two processes. Based on a visual analysis of the densities, it seems that the zero drift and pre-trained diffusion model initializations produce similar results around the observations, although the Schrödinger bridge initialization gave slightly sharper results at terminal time.

## D DIFFERENTIABLE RESAMPLING

In the ISB model steps ① and ③ presented in Sec. 3.1, we applied differentiable resampling (see Corenflos et al., 2021). Resampling itself is a basic block of particle filtering. A differentiable resampling step transports the particles and weights $(\tilde{\mathbf{x}}_{t_k}^i, w_{t_k}^i)$ to an uniform distribution over a set of particles through applying the *differentiable* ensemble transport map $\mathbf{T}_{(\varepsilon)}$, that is

$$(\tilde{\mathbf{x}}_{t_k}^i, w_{t_k}^i) \to (\tilde{\mathbf{X}}_{t_k}^\top \mathbf{T}_{(\varepsilon),i}, 1/N) = (\mathbf{x}_{t_k}^i, 1/N), \tag{49}$$

where $\tilde{\mathbf{X}}_{t_k} \in \mathbb{R}^{N \times d}$ denotes the stacked particles $\{\tilde{\mathbf{x}}_{t_k}^i\}_{i=1}^N$ at time $t_k$ before resampling and $\mathbf{x}_{t_k}^i$ denotes the particles post resampling. Here we give the definition of the map $\mathbf{T}_{(\varepsilon)}$ and review the regularized optimal transport problem which has to be solved to compute it. We partly follow the presentation in Sections 2 and 3 of Corenflos et al. (2021), but directly apply the notation we use for particles and weights and focus on explaining the transport problem rather than the algorithm used to solve it.

The standard particle filtering resampling step consists of sampling $N$ particles from the categorical distribution defined by the weights $\{w_{t_k}^i\}_{i=1}^N$, resulting in the particles with large weights being most likely to be repeated multiple times. A result from Reich (2013) gives the property that the random resampling step can be approximated by a deterministic ensemble transform $\mathbf{T}$. In heuristic terms, the ensemble transform map will be selected so that the particles $\{\mathbf{x}_{t_k}^i\}_{i=1}^N$ will be transported with minimal cost, while allowing all the weights to be uniform.

Let $\mu$ and $\nu$ be atomic measures, $\mu = \sum_{i=1}^N w_{t_k}^i \delta_{\tilde{\mathbf{x}}_{t_k}^i}$ and $\nu = \sum_{i=1}^N N^{-1} \delta_{\tilde{\mathbf{x}}_{t_k}^i}$, where $\delta_x$ is the Dirac delta at $x$. Then $\mu$ is the particle filtering distribution before resampling. Define the elements of a cost matrix $\mathbf{C} \in \mathbb{R}^{N \times N}$ as $C_{i,j} = \|\tilde{\mathbf{x}}_{t_k}^i - \tilde{\mathbf{x}}_{t_k}^j\|^2$, and the 2-Wasserstein distance between two atomic measures as

$$\mathcal{W}_2^2(\mu, \nu) = \min_{P \in S(\mu,\nu)} \sum_{i=1}^N \sum_{j=1}^N C_{i,j} P_{i,j}. \tag{50}$$

Above the optimal matrix $\mathbf{P}$ is to be found within $S(\mu, \nu)$, which is a space consisting of mixtures of $N$ particles to $N$ particles such that the marginals coincide with the weights of $\mu$ and $\nu$, formally

$$S(\mu, \nu) = \left\{ \mathbf{P} \in [0,1]^{N \times N} \mid \sum_{i=1}^N P_{i,j} = w_{t_k}^i, \sum_{j=1}^N P_{i,j} = \frac{1}{N} \right\}. \tag{51}$$

The entropy-regularized Wasserstein distance with regularization parameter $\varepsilon$ is then

$$\mathcal{W}_{2,\varepsilon}^2 = \min_{\mathbf{P} \in S(\mu,\nu)} \sum_{i=1}^N \sum_{j=1}^N P_{i,j} \left( C_{i,j} + \varepsilon \log \frac{P_{i,j}}{w_{t_k}^i \cdot \frac{1}{N}} \right). \tag{52}$$

The unique minimizing transport map of the above Wasserstein distance is denoted by $\mathbf{P}_\varepsilon^{\text{OPT}}$, and the ensemble transport map is then set as $\mathbf{T}_{(\varepsilon)} = N \mathbf{P}_\varepsilon^{\text{OPT}}$. This means that we can find the matrix $\mathbf{T}_{(\varepsilon)}$ via minimizing the regularized Wasserstein distance, which is done by applying the iterative Sinkhorn algorithm for entropy-regularized optimal transport (Cuturi, 2013).

