# OpenReview forum: "Transport with Support: Data-Conditional Diffusion Bridges"
_ICLR.cc/2023/Conference — Submitted to ICLR 2023_

### Official Review · Reviewer_aFhC · 2022-10-22

**Confidence:** 3
**Correctness:** 3
**Technical Novelty And Significance:** 4
**Empirical Novelty And Significance:** 2
**Recommendation:** 6

**Clarity, Quality, Novelty And Reproducibility:**

Clarity: See above

Quality and Novelty: I think leveraging particle filters to address the path-aware kind of Schrodinger bridge is a promising extension. Didn't check the proof.


Reproducibility: Although I like the insights proposed in De Bortoli (2021), the method alone is not that scalable. My question is that does this method extend to the likelihood training framework [1]?

[1] Likelihood Training of Schrödinger Bridge using Forward-Backward SDEs Theory


**Strength And Weaknesses:**


**Pros:**

1. This paper is proposed to address an interesting problem, a path-constrained Schrodinger bridge, where a particle filter kicks in to tackle the sparse observations;

**Cons:**

While I am not an expert in the field of particle filters, my biggest concern is the clarity:

1. more background introduction on particle filters may be needed for non-expert readers in the appendix;
2. what is the optimal transport map **$\boldsymbol{T}_{\varepsilon}$**;
3. why do we need the H-nearest neighbours?
4. maybe I am wrong, i think simulated annealing only proposed to gradually decrease the noise. Why does decreasing and then increasing the noise scale resemble simulated annealing?
5. the resampling steps may be detailed.

**Summary Of The Paper:**

The authors proposed the iterative smoothing bridge by leveraging ideas from Schr\"{o}dinger bridge and particle filter. Such a method allows us to learn constrained stochastic processes governed by sparse observations at intermediate stages. The algorithm is evaluated on synthetic data, real data, and small-scale high-dimensional data. This is an interesting problem and worth a deeper investigation and the proposed methodology seems to be a reasonable candidate.

**Summary Of The Review:**

An interesting method to solve an important problem, not clear/scalable enough though.

---

> ### Author Response · Authors · 2022-11-10
> **Response to reviewer aFhC**
>
> We thank the reviewer on their comments on the manuscript, and on their support on the problem we study as an interesting task. Below we address the questions and concerns in the order they were raised in the review.
>
> > Details on differential resampling
>
> We have included additional material on differentiable resampling in the supplementary material (see new App. D) of the revised version to help non-expert readers, and now include a reference to a recommended resource ([1]) on particle filtering fundamentals.
>
>
> > The optimal transport map
>
> The optimal transport map (or more specifically, the ensemble transform) $T_{\epsilon}$ is a $N \times N$ matrix which defines a map from the weighted particles $ (w_{i}, z_i )_{i=1}^N $
>
>  to uniformly weighted particles $ (1/N, T_{\epsilon} x_{i})_{i=1}^N $ in the optimal way, in terms of $\epsilon$ -regularized Wasserstein distance, see [2] Section 3 on entropy-regularized optimal transport for additional details.
>
> > Why do we need the H-nearest neighbours?
>
> We chose to use $H$-nearest neighbours to include local density information—such a choice allows us to take into account the density of observations instead of only the nearest observation. In experiments where the number of observations is high, letting $H$ grow and the observation noise to go down has meaningful theoretical properties, see Proposition 2 for a result on how the particle weights converge to the true underlying density with infinite data. We have added a remark on why letting $H>1$ is a meaningful choice in Section 3.2.
>
> > Why does decreasing and then increasing the noise scale resemble simulated annealing?
>
> It is indeed true that simulated annealing does not by itself include a noise schedule where the noise is increasing—the noise schedule we chose is a combination of the two desired properties: first simulated annealing (noise decreasing), and then convergence to the IPFP with no observations (noise increasing). We have improved the description of the heuristic in the manuscript (Sec. 3.2) to avoid confusion.
>
> > Scalability
>
> On scalability, we consider the method presented in [3] to still be reasonably scalable with regards to dimensionality. For future work with high-dimensional observational data, it is reasonable to assume that the high-dimensional observations are driven by a lower-dimensional latent process, thus resulting in a diffusion process running in a low-dimensional space.
>
> > Does this method extend to the likelihood training framework?
>
> Modifying the ISB model to work with the [4] likelihood-based training is a promising future direction; in that case the particles would have to be included not in the transition densities but rather in a continuous-time loss.
>
> **References**
>
> [1] Nicolas Chopin and Omiros Papaspiliopoulos. *An Introduction to Sequential Monte Carlo*. Springer, 2020.
>
> [2] Adrien Corenflos, James Thornton, George Deligiannidis, and Arnaud Doucet. Differentiable particle filtering via entropy-regularized optimal transport. In *Proceedings of the 38th International Conference on Machine Learning (ICML)*, volume 139 of Proceedings of Machine Learning Research, pp. 2100–2111. PMLR, 2021.
>
> [3] De Bortoli, V., Thornton, J., Heng, J. and Doucet, A., 2021. Diffusion Schrödinger bridge with applications to score-based generative modeling. *Advances in Neural Information Processing Systems*, 34, pp.17695-17709.
>
> [4] Evangelos A. Theodorou, Tianrong Chen, Guan-Horng Liu. Likelihood training of Schrödinger bridge using forward-backward SDEs theory. In *International Conference on Learning Representations (ICLR)*, 2022.

---

### Official Review · Reviewer_JvCw · 2022-10-24

**Confidence:** 2
**Correctness:** 3
**Technical Novelty And Significance:** 3
**Empirical Novelty And Significance:** 4
**Recommendation:** 6

**Clarity, Quality, Novelty And Reproducibility:**

Proposed problem is novel, well done experiment section, notation is not standard for stochastic calculus which makes it harder to read

**Strength And Weaknesses:**

Strength:
1. strong theoretical formulation of the proposed approach
2. well explained algorithm
3. well performed experiments

Weaknesses:
1. lack of rigorous proofs of main propositions (proof sketches of proposition 1 and 2)

**Summary Of The Paper:**

Authors present computationally efficient framework for learning data conditional diffusion bridges using Iterative Smoothing Bridge. The proposed framework is assessed by experimental results on both synthetic and real world data.


**Summary Of The Review:**

The paper is well written, presents novel framework with experimental assessment of it. Proofs of propositions on which the framework relies are not full but it seems that they are more or less true based on provided sketches.

---

> ### Author Response · Authors · 2022-11-10
> **Response to reviewer JvCw**
>
> We thank the reviewer for their supportive comments and appreciate they found our approach to be well explained and empirically justified.
>
> Regarding you concern related to the proofs: Our proofs of Propositions 1 and 2 (included in App. A.1 and App A.3) are sketches, but include a more detailed presentation than done in previous work (cf., [1], App. E.1) for related theoretical results.
>
> **References**
>
> [1] De Bortoli, V., Thornton, J., Heng, J. and Doucet, A., 2021. Diffusion Schrödinger bridge with applications to score-based generative modeling. Advances in Neural Information Processing Systems, 34, pp.17695-17709.

---

### Official Review · Reviewer_3a4i · 2022-10-30

**Confidence:** 3
**Clarity, Quality, Novelty And Reproducibility:** As listed above.
**Correctness:** 3
**Technical Novelty And Significance:** 3
**Empirical Novelty And Significance:** 3
**Recommendation:** 6

**Strength And Weaknesses:**

Strengths:
- They study an important problem: that of learning dynamical processes given some sparse observations at intermediate times

- The ISB method is novel

- They achieve some improved empirical results when compared against related methods in the literature

- The work is fairly well written

Weaknesses:
- While their method does offer some improvement compared to related methods, it is not a significant improvement, see for example table 2, figure 6, figure 7

- The authors give little theoretical justification for their method, beyond Prop. 3 which is not rigorously proved and in any case is essentially well-known.

- The core elements of the method appeared in prior works, namely De Bortoli et al 2021 and Corenflos et al 2021. The novelty of their method is in the combination of these approaches.

Writing feedback:
- Typos in background paragraph on page 3: should be $\mathcal{C} = C([0, T]; \mathbb{R}^d)$. There's an extra "to" in the sentence that begins with $x_t$
- The differentiable re-sampling procedure is hardly explained at all, and only a passing reference to Corenflos et al is given. Comprehension would be greatly aided by giving some discussion of this method, if only in an appendix. Also, it seems that the differentiability of the re-sampling method is not used in your method - is this correct?

De Bortoli, V., Thornton, J., Heng, J., & Doucet, A. (2021). Diffusion Schrödinger bridge with applications to score-based generative modeling. Advances in Neural Information Processing Systems, 34, 17695-17709.

Corenflos, A., Thornton, J., Deligiannidis, G., & Doucet, A. (2021, July). Differentiable particle filtering via entropy-regularized optimal transport. In International Conference on Machine Learning (pp. 2100-2111). PMLR.


**Summary Of The Paper:**

This paper studies the problem of learning a dynamical process given sparse observations of the process at intermediate times using a modification of the Schrodinger bridge process. In particular, the authors propose a method they term Iterative Smoothing Bridge (ISB) which alternates fitting forward and backward drifts parametrized by neural networks with a particle filtering and update step. This latter step is included to incorporate potentially sparse observations of the intermediate time dynamics. The authors discuss connections between their methods and stochastic control and give some theoretical claims. They then give an empirical study of their method.


**Summary Of The Review:**

While this work proposes a new method -- ISB -- for the important problem of learning dynamical processes with sparse observations, it does not achieve strong empirical performance and has limited conceptual novelty. For these reasons I think it is slightly too weak to merit acceptance.

---

> ### Author Response · Authors · 2022-11-10
> **Response to reviewer 3a4i**
>
> We thank the reviewer for their comments on the manuscript, and for their positive feedback on the importance of the problem we study and the novelty of our work.
>
> > Experimental validation
>
> On the experimental results presented in Table 2 and Figures 7 and 8 (new numbering in the revised version), we show clear performance improvement in using the intermediate data. The performance in terms of adhering to the terminal constraints had not improved, but we consider it behaviour as expected for the ISB model. As the ISB model has to take into account further constraints compared to IPFP, it acts as a compromise between the intermediate and terminal constraints. This is in-line with what one would expect form the methodology, and shows clear benefits of the proposed method.
>
> > Novelty
>
> The novelty of our method is both in the problem statement itself and the combination of iterative bridge methods and differentiable particle filtering. Earlier work on constrained stochastic control assumes that an analytically defined path constraint has been set, see for example [1]. To our knowledge, no work as general on solving such data-driven constrained problems exists within the machine learning community.
>
> The theoretical justification of our method is through two properties of the reversal of filtering densities: 1) As highlighted in the proof of Proposition 1 and in Appendix A.2, the learned reverse SDE coincides with the smoothing density, and 2) With abundant data and an initialization which covers the data generating distribution, the particle filtering weights match the true underlying density, see Proposition 2.
>
> > Details on differential resampling
>
> We are writing additional material on differential resampling in Appendix D of the revised manuscript, to ensure that our method is approachable to readers unfamiliar with the topic. We indeed do not differentiate over the particle filtering steps, so the differentiability of the resampling step is in that sense not necessary. Still, it is crucial for the proof of Proposition 1 that the resampling step acts as a linear map on the pre-sampling particles and weights, and the optimal transport map results in the particles moving minimally while maintaining the distribution set by the weights and particles.
>
> **References**
>
> [1] Dimitra Maoutsa and Manfred Opper. Deterministic particle flows for constraining stochastic nonlinear systems. arXiv preprint arXiv:2112.05735, 2021.

---

> > ### Comment · Reviewer_3a4i · 2022-11-23
> > **Thanks**
> >
> > Thanks for your thorough responses, I have updated my score accordingly.

---

### Official Review · Reviewer_V4yF · 2022-11-02

**Confidence:** 3
**Correctness:** 3
**Technical Novelty And Significance:** 2
**Empirical Novelty And Significance:** 3
**Recommendation:** 6

**Clarity, Quality, Novelty And Reproducibility:**

Clarity and Quality
- Sec. 3.1 is somewhat ambiguous. The authors need to take more effort on the writing to make the logic much clearer.
- The results of IPFP should be included in Fig. 3 for comparison.
- There are also some typos:
    - In the fourth-to-last line of page 4, $b_{l,\phi}$ should be $b_{l-1,\phi}$.
    - In the second line under equation (7), it should be $f_{l-1, \theta}$ and $g_{l+1, \phi}$
    - In the experiment of **Single-cell embryo RNA sequences**, why is the PCA used? What happens if the experiment is conducted on the original data?

Novelty
- The paper proposes an interesting problem, and the solution seems work.

Reproducibility
- Without source code, the work is hard to reproduce.

**Strength And Weaknesses:**

Strength
- The proposed problem may be important in many different applications with sparse intermediate observations, especially in the medical area.
- It is reasonable to use $L^2$ loss to handle the forward and backward drifts.
- Experimental results are convincing.

Weakness
- No convergence guarantee of the proposed method.
- For step 2 and 4, since there only exists sparse intermediate observations, to make the algorithm converge, it seems that a large number of samples is needed to make the method converge.
- It is reasonable to assume both $g$ and $\beta$ the same in both equation (5) and (6)?
- I may miss this part in the paper. Empirically, how to define $g$?
- The second paragraph of Step 1 in Sec. 3.1 is not very clear.

**Summary Of The Paper:**

In this paper, the authors propose to add sparse constraints to the original Schrodinger bridges through optimal control. Specifically, the paper assumes that there exist some intermediate sparse samples during the diffusion process. By modifying the Iterative Proportional Fitting procedure (IPFP) method with spare intermediate constraints, the Iterative Smoothing Bridge (ISB) method is proposed. Experiments show that the ISB method can help the forward and backward drift functions successfully evolve toward the intermediate observations.

**Summary Of The Review:**

Generally, the paper proposes an interesting problem. But the ambiguity in writing and implementation makes it hard to follow.

---

> ### Author Response · Authors · 2022-11-10
> **Response to reviewer V4yF**
>
> We thank the reviewer for their comments on the manuscript and their supportive remarks on its applicability and experimental results. We summarize how we have addressed the issues raised in the review below.
>
> > Clarity
>
> The main concerns relate to the clarity of the manuscript. We now provide additional material on differential resampling in a new section in the appendix (App. D), and for a comprehensive introduction to particle filtering, we now refer the reader to [1] in the Related Work (Sec. 1.1).
>
> > IPFP results for Figure 3
>
> This is a good idea. We have included a plot of the IPFP dynamics for the scikit-learn experiments in the revised manuscript (see Figure 6 in the appendix) that directly highlights the practical effect of including intermediate observations.
>
> > Reproducibility
>
> The supplementary material includes an example code implementation for a 2D scikit-learn experiment, and we commit to publishing code for the rest of the experiments once the manuscript is accepted.
>
> > Number of samples
>
> The number of sampled trajectories used while training the reverse drift networks in steps 2 and 4 was in most experiments set to 1000. This is not particularly high in a particle filtering context and is of the same order of magnitude as in earlier work on diffusion models. We have included the number of particles in the revised version, in the beginning of Section 4.
>
> > Single-cell embryo RNA experiment
>
> We chose to use PCA for dimensionality reduction in the single-cell experiment since earlier work [2] also uses it, and thus to make sure that the comparison is fair (comparing “apples to apples”). Studying more complex observation models by, for example, including the dimensionality reduction into the observation model itself is a promising future extension of our work and could further expand the applicability of ISB models. We have added a remark on PCA to the single-cell experiment section on page 9.
>
> **References**
>
> [1] Nicolas Chopin and Omiros Papaspiliopoulos. *An Introduction to Sequential Monte Carlo*. Springer, 2020.
>
> [2] Francisco Vargas, Pierre Thodoroff, Austen Lamacraft, and Neil Lawrence. Solving  Schrödinger bridges via maximum likelihood. *Entropy*, 23(9):1134, 2021.

---

> > ### Comment · Reviewer_V4yF · 2022-11-16
> > **Thanks and one more question**
> >
> > Thanks for the detailed clarify. For the last point, I am wondering if the algorithm can run on the original data manifold space, instead of the PCA space, which can be treated as an Euclidean space.

---

> > > ### Author Response · Authors · 2022-11-16
> > > **Running the single-cell experiment in non-Euclidean space**
> > >
> > > We limited our analysis to diffusion processes in Euclidean spaces as those models are typically used in the latent space. That said, we believe that our algorithm can be extended, with some technical modifications, to non-Euclidean spaces by leveraging recent advancements in the field, such as Bortoli et al. 2022 for Riemannian spaces or Campbell et al. for discrete data. We consider this an interesting direction for future research but believe it is outside the scope of this submission.
> > >
> > > [Bortoli et al. 2022] V. D. Bortoli, E. Mathieu, M. Hutchinson, J. Thornton, Y. W. Teh, and A. Doucet. Riemannian Score-Based Generative Modelling. To appear in NeurIPS 2022.
> > >
> > > [Campbell et al. 2022] A. Campbell, J. Benton, V. D. Bortoli, T. Rainforth, G. Deligiannidis, and A. Doucet. A continuous time framework for discrete denoising models. To appear in NeurIPS 2022.

---

### Author Response · Authors · 2022-11-10
**Response to all reviewers**

We thank all reviewers for their thoughtful and constructive comments on the manuscript. All reviewers considered the problem we study to be important and relevant to the ICLR community. Other strengths mentioned were our novel approach and the promising experimental results. The main concerns listed in the reviews relate to clarity and the reviews included various good suggestions to improve presentation.

We have now revised our manuscript and supplement. A comprehensive list of the changes is provided in the individual replies to each reviewer. The main changes were:
* Adding a new Fig. 6 to the supplement that shows the IPFP equivalent for Fig. 3
* Adding background material on differential resampling (new App. D)
* Clarifying the phrasing in the methods (Sec. 3) section that were raised by the reviewers
* Clarifying and justifying details in the experiments (Sec. 4) that were raised by the reviewers

---

### Decision · Program_Chairs · 2023-01-20

**Decision:**

Reject

**Justification For Why Not Higher Score:**

the paper is not yet ready for publishing due to lack of clarity

**Justification For Why Not Lower Score:**

N/A

**Metareview: Summary, Strengths And Weaknesses:**

The paper proposes the  Iterative Smoothing Bridge (ISB) where sparse observations are known on the trajectories for transporting a source to a target distribution. The paper proposes to combine learned diffusion models with particles filtering to achieve this. The paper is promising and the idea of trajectory inference in optimal transport is interesting nevertheless the paper lacks a lot on the clarity.

After reading the paper multiple times it is not fully clear how the method is implemented and how  the main  building blocks are connected. Authors added few clarifications in the appendix during the rebuttal  but the main paper is still not easy to follow. We suggest the authors to have a more transparent presentation of the proposed method with an algorithm summarizing the steps and the main computations performed.



**Summary Of Ac-Reviewer Meeting:**

N/A